# DELE1 tracks perturbed protein import and processing in human mitochondria

Evelyn Fessler 1✉, Luisa Krumwiede 1 & Lucas T. Jae 1✉

Protein homeostatic control of mitochondria is key to age-related diseases and organismal decline. However, it is unknown how the diverse types of stress experienced by mitochondria can be integrated and appropriately responded to in human cells. Here we identify perturbations in the ancient conserved processes of mitochondrial protein import and processing as sources of DELE1 activation: DELE1 is continuously sorted across both mitochondrial membranes into the matrix and detects different types of perturbations along the way. DELE1 molecules in transit can become licensed for mitochondrial release and stress signaling through proteolytic removal of N-terminal sorting signals. Import defects that occur at the mitochondrial surface allow DELE1 precursors to bind and activate downstream factor HRI without the need for cleavage. Genome-wide genetics reveal that DELE1 additionally responds to compromised presequence processing by the matrix proteases PITRM1 and MPP, which are mutated in neurodegenerative diseases. These mechanisms rationalize DELE1-dependent mitochondrial stress integration in the human system and may inform future therapies of neuropathies.

¹ Gene Center and Department of Biochemistry, Ludwig-Maximilians-Universität München, Feodor-Lynen-Strasse 25, 81377 Munich, Germany.
✉email: fessler@genzentrum.lmu.de; jae@genzentrum.lmu.de

M itochondrial dysfunction is a hallmark of aging and severe age-related diseases, ranging from heart failure, to cancer, to different forms of neurodegeneration, such as Parkinson's disease, Alzheimer's disease, or amyotrophic lateral sclerosis[1,2]. Mitochondrial fidelity depends on the homeostatic control of the mitochondrial proteome, which is complicated by the evolutionary origin of the organelle. As relatives of alphaproteobacteria, mitochondria need to carefully coordinate gene expression from their own genome (mtDNA) with that of the nuclear genome[2,3]. Additionally, the import of nuclear-encoded mitochondrial proteins requires their translocation across one or two membranes in an unfolded state and is accompanied by extensive proteolytic processing in many cases[4]. In the light of these delicate tasks, it is remarkable that mitochondria do not encode any stress response genes[1] and instead need to alert the surrounding cell about threats to mitochondrial homeostasis. This precipitated complex stress sensors and signaling cascades, most of which have been deciphered in landmark studies employing model organisms like nematodes and yeast[5–9]. However, how mitochondrial stress is sensed and signaled in the human system remains substantially less understood[10].

Through genome-wide phenotypic genetic screening, recently, we and others identified a pathway that relays mitochondrial perturbations to the cytosol in human cells: DELE1, which localizes to mitochondria in the steady-state, is a substrate of the stress-activated mitochondrial protease OMA1. Its C-terminal cleavage fragment (S-DELE1) subsequently moves to the cytosol by an unknown mechanism. There, it binds and activates the kinase HRI, which phosphorylates eIF2α, setting off the integrated stress response (ISR). This can be beneficial or detrimental, depending on the context of the mitochondrial insult[11–14]. Despite the important role of this pathway in the human mitochondrial stress response, little is known about its central factor DELE1 and how it can mechanistically accommodate its function as a stress relay. Here, we reveal DELE1 to be a key integrator of different types of perturbations in the ancient procedures of mitochondrial protein import and processing via its own flux through these systems. We show that DELE1 itself is a client of the presequence import pathway, reminiscent of the import stress sensing transcription factor ATFS-1 in the nematode *Caenorhabditis elegans*[5]. Different perturbations of protein import across the inner mitochondrial membrane (IMM) into the matrix result in the release of newly synthesized DELE1 precursors en route to this compartment. This mechanism involves the removal of N-terminally encoded DELE1 sorting signals by OMA1. We find that mimicking this shortening of DELE1 by a synthetic protease cleavage event is sufficient for its release into the cytosol and activation of the ISR. DELE1 can also alert the cell about perturbations affecting the pores of the outer mitochondrial membrane (OMM) independently of OMA1 by engaging HRI in its immature precursor form (L-DELE1). Finally, we demonstrate that in addition to import defects, DELE1 recognizes glitches in the proteolytic processing of mitochondrial precursor proteins. These can arise from mutations in the mitochondrial proteases MPP and PITRM1, which cause forms of neurodegeneration and cognitive defects and have been linked to Alzheimer's pathogenesis[15–17]. Together, these results position DELE1 as a multi-modal relay of mitochondrial import and precursor processing stress in the human system and point to a critical role of its activities in certain settings of human disease.

## Results

### Genome-wide screen for regulators of DELE1. While DELE1[HA] expressed from its endogenous locus can be detected by hemagglutinin (HA) immunoblotting using whole-cell lysates[11], we noticed that the protein is very short-lived in the steady-state, as it rapidly disappeared when protein synthesis was blocked with cycloheximide (CHX) (Fig. 1a). This suggests that the human cell has implemented mechanisms that keep DELE1 in check in the absence of mitochondrial stress, in line with its potentially dangerous ability to trigger the ISR[11,12]. To unbiasedly investigate the factors that control the fate of DELE1, we sought to carry out a genome-wide phenotypic screen in HAP1 cells[18]. Due to the lack of antibodies that specifically recognize DELE1[12], we utilized CRISPR-Cas9 to engineer the endogenous *DELE1* locus in haploid HAP1 cells with a C-terminal ALFA-tag[19] (Fig. 1b). The ALFA-tag is recognized by a high-affinity nanobody, which resulted in a specific signal for endogenous DELE1 in flow cytometry despite its overall modest protein levels (Fig. 1c). Importantly, as we had previously demonstrated for endogenous DELE1[HA][11], the endogenous ALFA-tagged protein is also functional, evident by the ability of S-DELE1[ALFA] to co-precipitate with HRI in the context of mitochondrial stress elicited by the depolarizing ionophore carbonyl cyanide *m*-chlorophenyl hydrazone (CCCP) (Fig. 1d, e).

Next, we subjected the ALFA-engineered HAP1 cells to ultradeep random genome mutagenesis by gene-trap[18] and analyzed mutations in cells with high or low DELE1 signal by deep sequencing (Supplementary Fig. 1a). Cells with low DELE1[ALFA] signal were strongly enriched for mutations in *DELE1* itself, genetically corroborating the validity of the ALFA-tag genome engineering strategy (Fig. 1f; Supplementary Data 1). Moreover, decreased DELE1 signal was also observed in cells with gene-trap mutations in the RNA-binding factor *CLUH* ($P_{adj} < 7.52 \times 10^{-46}$), rationalizing our prior observation that induction of the ISR transcription factor CHOP after mitochondrial stress depends on CLUH[11]. CLUH has been reported to specifically bind the transcripts of nuclear-encoded mitochondrial proteins and aid their stability and translation[20]. In its absence, these proteins are depleted from the mitochondrial proteome and mitochondrial morphology is profoundly altered as a result of defective mitophagy[21]. Notably, the DELE1 mRNA can be found among transcripts that are significantly enriched in CLUH RNA immunoprecipitation experiments[22].

The genetic screen also identified a number of mitochondrial proteases[23], including CLPP and XPNPEP3, as well as AFG3L2, PARL and OSGEPL1. Most strikingly, among cells with increased DELE1 signal, mutations were highly enriched in the mitochondrial presequence protease PITRM1 ($P_{adj} < 1.56 \times 10^{-10}$). PITRM1 was first identified in plants as an oligopeptidase in charge of degrading targeting sequences for chloroplasts and mitochondria[24]. In the human system, PITRM1 degrades the presequences of mitochondrial precursor proteins after their release from the precursor by the activity of the matrix-resident mitochondrial processing peptidase (MPP)[25]. Improper activity of PITRM1 results in the accumulation of presequences and incompletely processed mitochondrial precursor proteins, which causes a perturbation in the cellular proteome. It was shown that this involves feedback inhibition of MPP[16,26]. Clinically, PITRM1 defects cause progressive forms of neurodegeneration[27–29]. Notably, PITRM1 can also degrade other short peptides, including Aβ species which accumulate in the brains of Alzheimer's disease patients[15] and its activity was found to counteract Alzheimer's pathology in a mouse model[30]. Mechanistically, it has been proposed that Aβ peptides overwhelm the processing capacity of PITRM1 and thus result in presequence accumulation[16,31].

To confirm the roles of CLUH and PITRM1 as regulators of DELE1, we first exposed HAP1 *DELE1[ALFA]* cells to Cas9 and sgRNAs directed against these genes, which effectively depleted CLUH or PITRM1 proteins in the respective polyclonal populations (Fig. 1g). As predicted by the genetic screen,

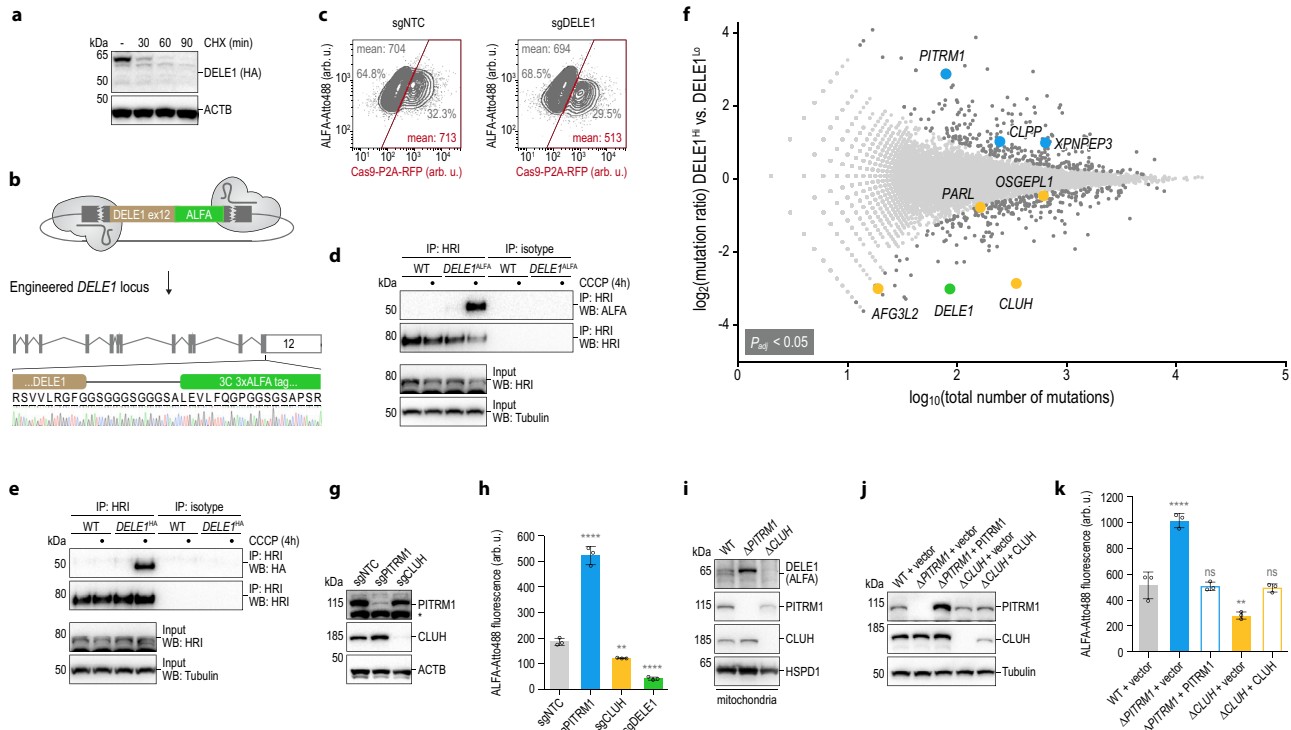

**Fig. 1 Genome-wide screen for regulators of DELE1. a** HeLa cells expressing endogenous *DELE1*[HA] were treated with cycloheximide (CHX) and analyzed by immunoblotting. ACTB, β-actin. **b** The endogenous *DELE1* locus in HAP1 cells was engineered with a C-terminal triple ALFA tag using CRISPR-Cas9. In-frame fusion verified by Sanger sequencing. **c** HAP1 *DELE1*[ALFA] cells were lentivirally transduced to express a single guide non-targeting control (sgNTC) or an sgRNA targeting *DELE1* (sgDELE1) together with Cas9-P2A-RFP at a multiplicity of infection < 1, such that each population comprises sgRNA-containing cells (RPF+) and non-transduced wild-type cells. DELE1[ALFA] was stained with Atto488, values specify mean Atto488 intensity in transduced (RFP+; red box) and non-transduced (RFP-) cells. **d**, **e** HAP1 cells of the indicated genotypes were treated with CCCP and the ability of endogenous HRI to co-precipitate the respective S-DELE1 proteins was analyzed by immunoblotting. WT, wild-type. **f** Haploid genetic screen for regulators of DELE1. Per gene (dots), the ratio of the frequency of mutations in DELE1[ALFA]-high versus DELE1[ALFA]-low cells (*y*-axis) is plotted against the combined number of unique mutations identified in both populations (*x*-axis). Genes significantly enriched for mutations in either population are dark gray or colored (two-sided Fisher's exact test, FDR-corrected *P*-value ($P_{adj}$) < 0.05). **g** HAP1 *DELE1*[ALFA] cells were exposed to sgRNAs as indicated. Knockout efficiency was analyzed by immunoblotting. Asterisk: nonspecific band. **h** DELE1 protein levels (cells as in (**g**)) analyzed by flow cytometry. Mean ± s.d. of *n* = 3 independent biological samples (arb. u., arbitrary units). Statistical significance compared to sgNTC assessed using ordinary one-way ANOVA with Dunnett's multiple comparisons correction. ****$P$ < 0.0001, **$P$ = 0.0094. **i** Mitochondrial DELE1 protein levels of clonal HAP1 *DELE1*[ALFA] PITRM1 or CLUH knockout cells analyzed by immunoblotting. **j** Clonal HAP1 *DELE1*[ALFA] PITRM1 and CLUH knockout cells reconstituted with the respective cDNAs or vector control analyzed by immunoblotting. **k** DELE1 protein levels (cells as in (**j**)) analyzed by flow cytometry. Mean ± s.d. of *n* = 3 independent biological samples. Statistical significance compared to WT + vector assessed with ordinary one-way ANOVA and Dunnett's multiple comparisons correction; ns, non-significant. ****$P$ < 0.0001, **$P$ = 0.0020, ns ≥ 0.9783.

DELE1 signal was increased in sgPITRM1-treated cells, whereas cells exposed to sgRNAs targeting CLUH showed decreased DELE1 levels (Fig. 1h; Supplementary Fig. 1b). This phenotype was also apparent at the level of mitochondria in cellular fractionation experiments (Fig. 1i; Supplementary Fig. 1c) and could be reverted by complementation of clonal PITRM1 knockout cells or CLUH knockout cells with the respective cDNAs (Fig. 1j, k). Together, these experiments demonstrate that DELE1 is controlled by positive and negative regulators in the steady-state, including such that are implicated in mitochondrial protein translation, sorting and maturation processes.

**Sorting and processing of DELE1.** These observations prompted us to investigate the mitochondrial import and processing of DELE1. The mitochondria targeting sequence (MTS) of DELE1 has been reported to encompass the N-terminal 100[12] or 115[32] amino acids, deletion of which indeed causes constitutive localization of DELE1 to the cytosol in either case[11]. Although unusually long in the case of DELE1, such N-terminal MTS signals are typically made up of a presequence, which adopts the

configuration of an amphipathic α-helix and is recognized by the TOM and TIM complexes, as well as other mitochondrial import components[4,33]. The presequence is released from the precursor protein by MPP during or after import[4,25], followed by its degradation by PITRM1[24]. Of note, both of these proteolytic events take place in the mitochondrial matrix.

Given the identified regulatory role of PITRM1 exerted on DELE1, we wondered whether DELE1 might be a client of the mitochondrial presequence pathway. To this end, we first analyzed DELE1 species generated in wild-type or OMA1-deficient cells in the steady-state and in the context of mitochondrial perturbation with CCCP or oligomycin (OM). This revealed that besides full-length DELE1 (L-DELE1) and its short version (S-DELE1), which is generated by OMA1 after mitochondrial stress[11,12], endogenous DELE1 gives rise to additional species, most notably an intermediate-sized protein residing in the mitochondrial fraction, which we term M-DELE1 (Fig. 2a; Supplementary Fig. 2a). We observed very similar results, including the generation of M-DELE1, when DELE1 was expressed from a cDNA (Supplementary Fig. 2b), demonstrating that

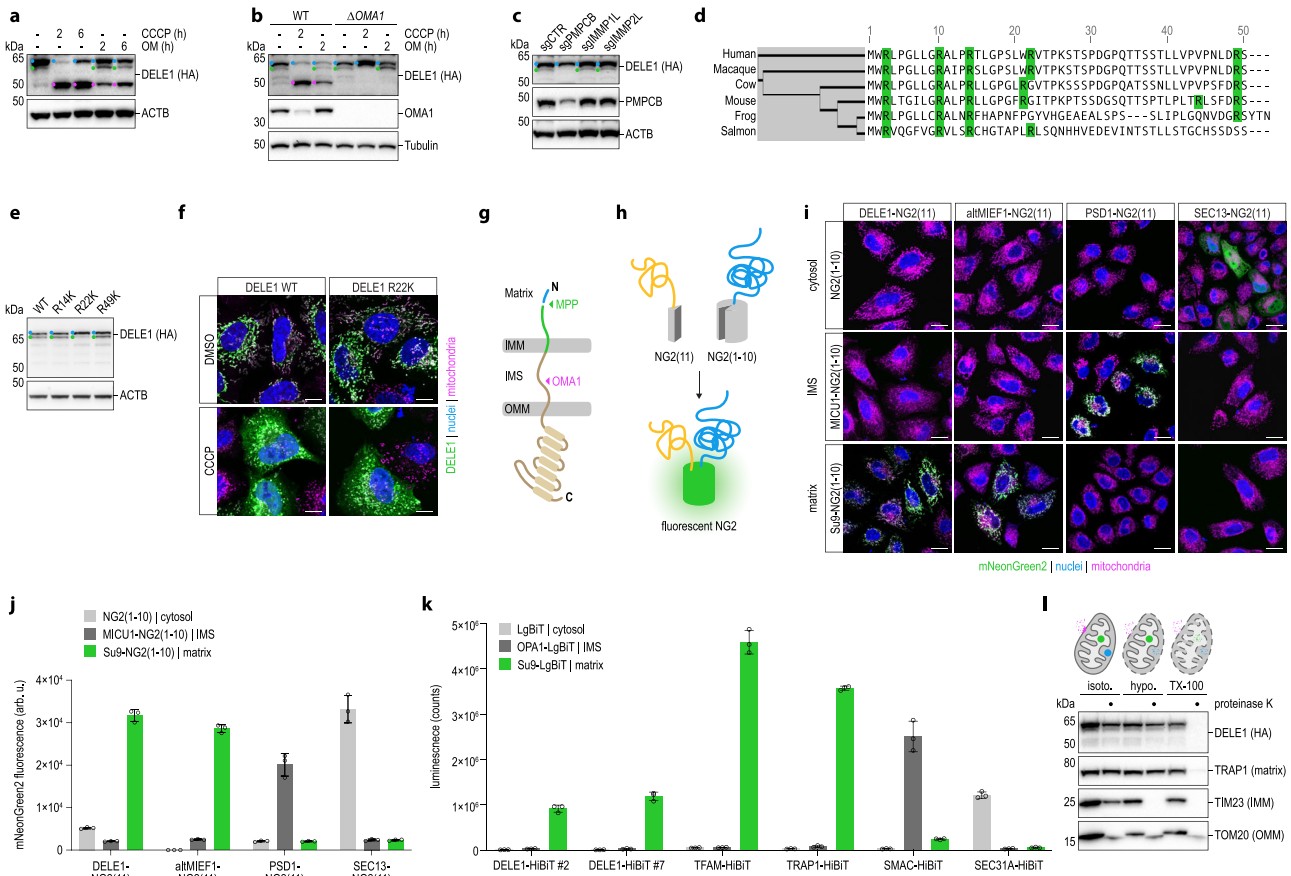

**Fig. 2 DELE1 is a substrate of MPP that can be fully imported into the mitochondrial matrix. a** HeLa *DELE1*[HA] cells treated as indicated and analyzed for endogenous DELE1 protein by immunoblotting. OM, oligomycin. Blue, L-DELE1. Green, M-DELE1. Pink, S-DELE1. **b** 293 T *DELE1*[HA] WT and OMA1 knockout (*ΔOMA1*) cells treated as indicated and analyzed by immunoblotting. **c** HeLa *DELE1*[HA] cells exposed to sgRNAs targeting the indicated genes for six days and analyzed by immunoblotting. **d** Alignment of the first 50 amino acids of DELE1 from different species. Arginines highlighted (green). **e** Wild-type DELE1 or the indicated arginine to lysine mutants transiently expressed in HeLa cells. Cell lysates analyzed by immunoblotting. **f** Confocal microscopy of HeLa cells transfected with indicated DELE1 variants. Scale bars,10 μm. Nuclei (DAPI, blue), mitochondria (MitoTrackerRed, pink), DELE1 (HA, green). **g** Schematic illustrating DELE1 import into mitochondria and its processing by MPP and OMA1. IMM, inner mitochondrial membrane. IMS, intermembrane space. OMM, outer mitochondrial membrane. **h** Principle of split mNeonGreen2 (NG2) complementation by fusion of the small (NG2(11)) and the large (NG2(1-10)) component to different proteins (yellow, blue) sorted into the same subcellular compartment. **i** Localization of DELE1 in HeLa cells analyzed via split NG2 system by confocal microscopy. NG2(11) was C-terminally fused to DELE1, altMIEF1, SEC13 or the sorting signal of PSD1 and co-expressed with NG2(1-10) directed to different subcellular compartments by fusion with the sorting signals of MICU1 or Su9. Scale bars, 20 μm. Nuclei (DAPI, blue), mitochondria (MitoTrackerRed, pink), mNG2 (green). **j** NG2 fluorescence as in (**i**) measured by flow cytometry. Mean ± s.d. of *n* = 3 independent biological samples. (**k**) Localization of endogenous DELE1 assessed by split luciferase reporter system. HeLa cells expressing the indicated endogenous proteins tagged with the small luciferase fragment (HiBiT) were transfected with the large luciferase fragment (LgBiT), unmodified or fused to the sorting signals of OPA1 or Su9. Luminescence measured 24 h post-transfection. Mean ± s.d. of *n* = 3 independent biological samples. **l** Accessibility of proteins to proteinase K in mitochondria isolated from HeLa *DELE1*[HA] cells in isotonic buffer (isoto.), upon osmotic swelling (hypotonic buffer, hypo.) or lysis of mitochondria (TX-100).

that M-DELE1 does not result from alternative splicing and that exogenous DELE1 closely recapitulates the behavior of endogenous DELE1 when expressed from a weak promoter. Importantly, unlike S-DELE1, the generation of M-DELE1 did not depend on OMA1 and was virtually abrogated in the presence of CCCP, but persistent in cells treated with OM (Fig. 2b; Supplementary Fig. 2b). CCCP depolarizes the IMM, which interferes with TIM23-dependent protein import into the matrix[34–36]. OM inhibits the ATP synthase, which depletes mitochondrial ATP and can hyperpolarize the IMM[37]. While ATP is needed for the complete import of precursors into the matrix, initial translocation of presequences across the IMM occurs in an electrophoretic fashion that does not require ATP[4,38–40]. The observed behavior of DELE1 thus suggested that M-DELE1 might be produced by an MPP-mediated cleavage event in the mitochondrial matrix.

To test this hypothesis, we monitored the generation of M-DELE1 in cells exposed to an sgRNA targeting the catalytic subunit of MPP—*PMPCB* in humans—or either subunit of the inner membrane peptidase (IMP, encoded by *IMMP1/2L* in humans), which functions in the release of several presequence proteins into the intermembrane space (IMS)[41,42]. While treatment with sgRNAs directed against IMP had no discernable effect on DELE1 processing, M-DELE1 was virtually lost in polyclonal cells exposed to sgRNAs targeting MPP, despite the fact that MPP depletion was inefficient, presumably because of its essentiality[43] (Fig. 2c). MPP cleaves presequences following so-called arginine rules[44–46]. Based on these rules, arginine conservation (Fig. 2d), and the observed molecular weight of M-DELE1, we speculated that R14, R22 and/or R49 of DELE1 might direct its processing by MPP. Whereas mutating R14 or

R49 to lysines did not noticeably affect the band pattern of DELE1, generation of M-DELE1 was blunted in the R22K mutant, phenocopying the effects observed with MPP-deficiency (Fig. 2e). Importantly, this effect was not caused by aberrant sorting of DELE1(R22K), as the mutant localized to mitochondria in the steady state and accumulated in the cytosol in the context of mitochondrial stress, indistinguishably from wild-type DELE1 (Fig. 2f). In line with these data, we also observed that the most N-terminal portion of the DELE1 MTS by itself promoted mitochondrial import of mCherry (Supplementary Fig. 2c). Together, these findings indicate that DELE1 contains a short cleavable presequence, removal of which is directed by MPP, and therefore presumably follows the presequence pathway towards the mitochondrial matrix (Fig. 2g).

To investigate a possible matrix localization of DELE1, we employed a combination of cell biological and biochemical approaches. First, we generated a series of split mNeonGreen2(1-10; NG2(1-10))[47] constructs targeted to different cellular compartments such that co-expression with a query protein tagged with the small NG2(11) fragment would result in fluorescence when both proteins meet in the same cellular compartment (Fig. 2h). Co-expression of DELE1-NG2(11) with cytosolic (no sorting signal), IMS-localized (MICU1), or matrix-localized (Su9) NG2(1-10) only resulted in strong fluorescence in the latter case. This pattern matched that of the matrix-sorted protein altMIEF1[48], but not that of IMS-localized PSD1 or cytosolic SEC13, which only complemented with NG2(1-10) targeted to their respective compartments (Fig. 2i, j). Next, we sought to extend these observations to endogenous DELE1. Due to its modest protein levels, we employed a split luciferase approach to enzymatically amplify potentially weak signals. We C-terminally tagged endogenous DELE1 with the small (11 amino acids) fragment of split luciferase and subsequently co-expressed the large luciferase fragment targeted to different mitochondrial subcompartments by suitable MTS fusions. Also in this assay, the pattern of DELE1 luminescence was most similar to that of other matrix proteins (Fig. 2k).

Finally, we set out to corroborate these genetic approaches orthogonally with biochemistry. To this end, we monitored degradation of DELE1 by proteinase K in the context of intact mitochondria, mitochondria in which the OMM was opened by hypotonic swelling, or such that were fully permeabilized with detergent (Fig. 2l; Supplementary Fig. 2d). As expected, in this assay the cytosol-exposed OMM protein TOM20 was cleaved in all conditions, whereas the matrix protein TRAP1 could only be cleaved when mitochondria were permeabilized with detergent. The IMM protein TIM23 was efficiently degraded when the OMM was impaired. Unlike for TIM23, much of the DELE1 signal remained intact under hypotonic conditions, but was completely lost with detergent, reminiscent of the behavior of TRAP1. However, in contrast to TRAP1, a fraction of DELE1 was also sensitive to proteinase K when the matrix was intact (isotonic or hypotonic conditions). This suggests that in the steady-state, a proportion of DELE1 can be found in the import trajectory, in line with its continuous de novo synthesis (Fig. 1a). Altogether, these observations indicate that DELE1 is targeted to the mitochondrial matrix via the presequence pathway in the steady-state and that the entire protein including its C-terminus can be sorted into this compartment.

**Mitochondrial release of DELE1.** To our knowledge, aside from the permeabilization of mitochondria during apoptosis[49], no general export mechanism for intramitochondrial proteins has been identified, and retro-translocation phenomena are limited to a small number of highly specialized factors[50–53]. Thus, the question arises how S-DELE1 is released to the cytosol upon mitochondrial perturbation. We and others have previously shown that rapid mitochondrial release of S-DELE1 depends on the mitoprotease OMA1, indicating that DELE1 does not merely accumulate in the cytosol because it can no longer enter into perturbed mitochondria[11,12]. In line with this, in OMA1-deficient cells treated with CCCP, virtually all of the mitochondrial DELE1 was readily accessible to proteinase K even in intact mitochondria, indicating an arrest early in its import trajectory (Fig. 3a). For PINK1, a mechanism has been identified by which newly synthesized precursor molecules are ejected from mitochondria upon cleavage by the IMM-resident protease PARL in the steady-state[50,51]. This prompted us to investigate, whether—analogously—newly synthesized DELE1 in the process of import constitutes the substrate of activated OMA1. We and others had previously observed that simultaneous treatment of cells with the protein synthesis inhibitor CHX plus a mitochondrial stressor that triggers OMA1 is permissive for the generation of some S-DELE1[11,12], likely due to the residual amount of immature DELE1 still in the process of import and thus accessible to IMS-resident OMA1. To eliminate this pool of DELE1, we briefly pretreated cells expressing endogenous DELE1[HA] with CHX, followed by a short pulse of CCCP (in the presence of CHX) and measured their ability to generate M-DELE1 and S-DELE1. Under these conditions, CHX pretreatment preferentially decreased the levels of immature L-DELE1 over M-DELE1 and greatly diminished the appearance of OMA1-generated S-DELE1 (Fig. 3b). Similar results were obtained when DELE1 was exogenously expressed from a cDNA (Fig. 3c). Additionally, when S-DELE1 was eliminated from the equation by means of OMA1 knockout, we observed that in the context of mitochondrial depolarization with CCCP, the equilibrium between L- and M-DELE1 is virtually fully shifted towards the full-length precursor (Supplementary Fig. 3a). Together, this demonstrates that newly synthesized immature DELE1 constitutes the main substrate for stress-activated OMA1.

Because the catalytic portion of OMA1 faces the IMS[54], we hypothesized that DELE1 molecules which are currently in the process of import are cleaved by OMA1 in the IMS, resulting in the removal of their sorting signals and the release of the C-terminal S-DELE1 fragment from mitochondria. To test this model, we created chimeric DELE1 proteins in which its MTS was replaced with heterologous MTS signals for targeting into the IMM/IMS (MICU1) or matrix (Su9) (Fig. 3d). We then expressed wild-type DELE1 or the chimeras and activated OMA1 either by CCCP, which compromises TIM23-dependent presequence transport into the matrix (and thus presequence processing by MPP) or OM, which is permissive for the initial steps of presequence translocation across the IMM. In the context of CCCP, wild-type DELE1 and all chimeric proteins gave rise to S-DELE1, in line with interference with IMM translocation and cleavage occurring in the IMS (Fig. 3e; Supplementary Fig. 3b). Strikingly, when OMA1 was activated with OM, only wild-type DELE1 and MICU1-DELE1, but not Su9-DELE1 could be cleaved into S-DELE1 (Fig. 3e). This indicates that in the setting of a polarized IMM, DELE1 cleavage by OMA1 in the IMS is productive for the natural DELE1 MTS or when DELE1 is artificially targeted to the IMM/IMS. By contrast, replacing the DELE1 MTS with the effective matrix-targeting signal of ATP synthase subunit 9 (Su9)[55] bypasses its cleavage by OM-triggered OMA1. These results also provided an opportunity to test if DELE1 can be released from mitochondria while its OMA1 cleavage site is located in the IMS but no longer once it has been sorted into the matrix. Expectedly, all constructs localized to mitochondria in unperturbed cells and accumulated in the cytosol when cells were treated with CCCP, in line with the observed

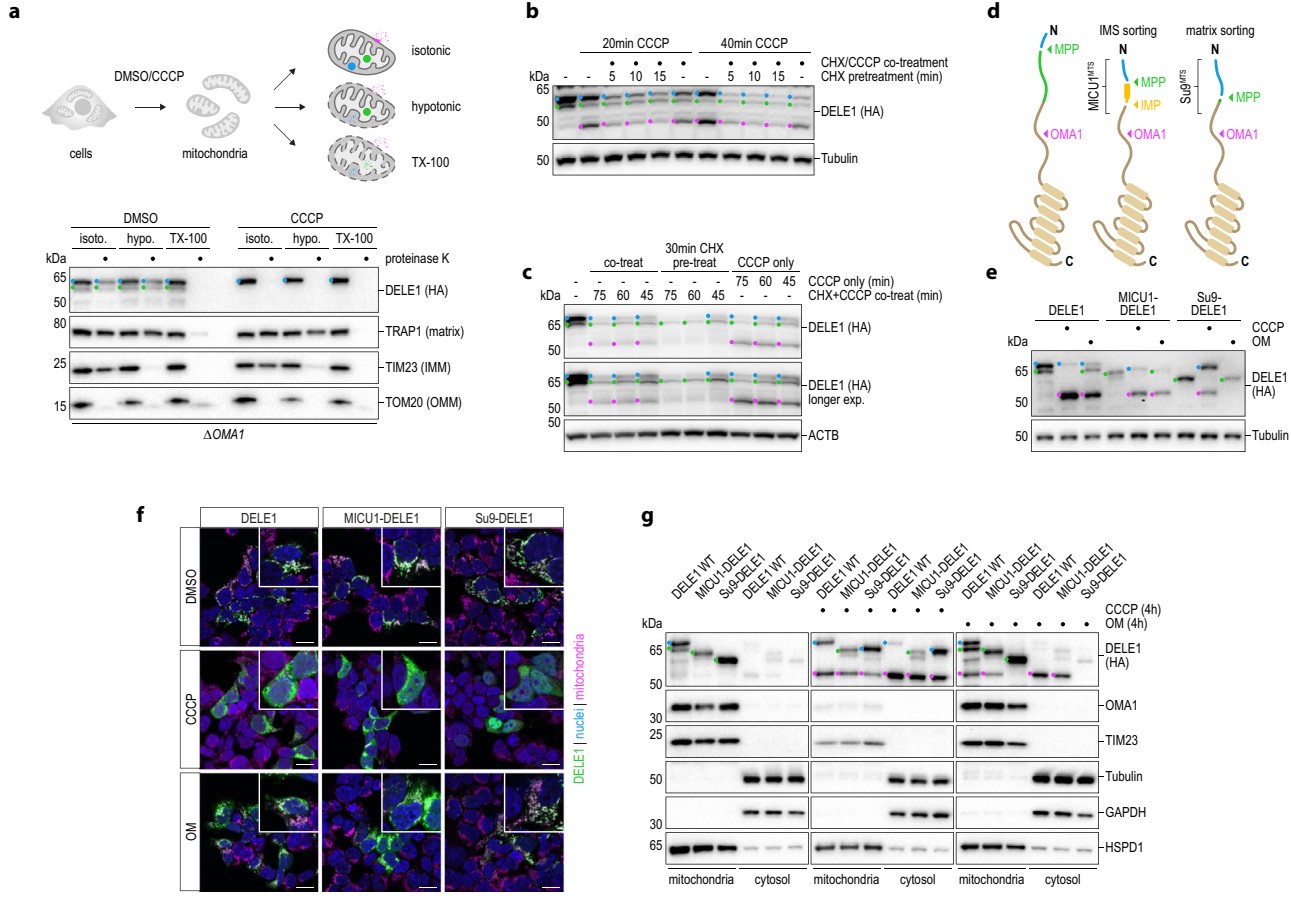

**Fig. 3 Release of DELE1 into the cytosol is coupled to sorting and maturation. a** Mitochondria isolated from 293T $DELE1^{HA}$ OMA1 knockout ($\Delta OMA1$) cells treated with DMSO or CCCP for 4 h were processed as in Fig. 2l. **b**, **c** HeLa $DELE1^{HA}$ cells (**b**) and HeLa cells stably expressing DELE1-HA (c) were pretreated with CHX to deplete newly synthesized DELE1 protein, followed by co-treatment of CHX with CCCP for the indicated times and analysis of DELE1 protein by immunoblotting. CCCP only and simultaneous CHX + CCCP treatment without CHX pretreatment serve as controls. **d** Schematic comparing wild-type DELE1 and chimeric DELE1 proteins carrying a heterologous MTS and their associated protease cleavage sites, used in the subsequent panels. **e** 293T DELE1 knockout cells stably expressing the indicated DELE1 variants were treated with CCCP or OM for 6 h. Processing of the DELE1 proteins was analyzed by immunoblotting. **f** 293T DELE1 knockout cells were transiently transfected with the specified DELE1 chimeras, treated for 4 h as indicated and analyzed by confocal microscopy. Scale bars, 20 μm. Nuclei (DAPI, blue), mitochondria (MitoTrackerRed, pink), DELE1 (HA, green). **g** Subcellular fractionation reveals the localization of different DELE1 species to mitochondria or cytosol upon CCCP or OM treatment in 293T DELE1 knockout cells transiently transfected with the indicated variants.

OMA1-mediated cleavage under these conditions. However, in the context of OM treatment, cytosolic accumulation of DELE1 was only apparent for wild-type DELE1 and MICU1-DELE1, whereas Su9-DELE1 remained mitochondrial (Fig. 3f). The differential release behavior was also mirrored in the ability of the chimeric constructs to induce the ISR marker CHOP (Supplementary Fig. 3c). To identify which DELE1 species exist in the respective compartments in the context of these perturbations, we next performed cellular fractionations. In line with the notion that the highly effective Su9 MTS can partially withstand import perturbations[35], Su9-DELE1 was mostly localized to mitochondria in its matured state but virtually undetectable in the cytosolic fraction of OM-treated cells. By contrast, cytosolic signal was readily observed for wild-type DELE1 and MICU1-DELE1 and largely consisted of S-DELE1 rather than precursor species (Fig. 3g). Similar results were obtained in cells treated with CCCP for these two proteins, whereas Su9-DELE1 displayed a substantial amount of cytosolic signal in the immature precursor state when mitochondria were depolarized. Together, this may indicate that, unlike the Su9 MTS, the endogenous MTS of DELE1 promotes cleavage by OMA1 possibly through association with early mitochondrial

destinations in the context of stress, rationalizing the observed mitochondrial retention of DELE1 in OMA1-deficient cells[11,12].

**Cleavage of DELE1 in the IMS is sufficient for its release from mitochondria.** This prompted us to investigate whether the role of OMA1 in the mitochondrial release of DELE1 is limited to severing the extended N-terminal portion containing the sorting signals. We had previously observed that cristae remodeling by the OMA1 substrate OPA1 was dispensable for DELE1 release[11]. To solve the question whether OMA1 serves additional functions in the release process, we sought to test whether the requirement for OMA1 in DELE1 release from mitochondria could be bypassed by a synthetic proteolytic event mediated by an unrelated protease. To this end, we functionalized the fast and effective[56] human rhinovirus 3C protease (3CP)[57] for use in human mitochondria by expressing a codon-optimized version targeted to different mitochondrial compartments alongside a DELE1 variant carrying a 3CP cleavage site (3CS) between the DELE1 N-terminus and C-terminal TPR segment. We reasoned that TOM20 (OMM) and TIM50 (IMS) fusions would encounter imported DELE1 at the TOM and TIM complex, respectively,

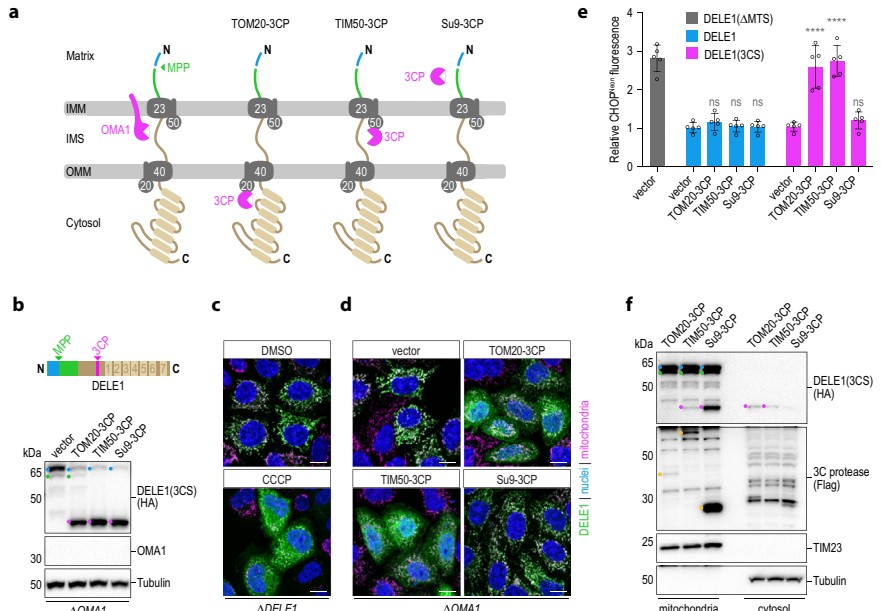

**Fig. 4 Cleavage of DELE1 in the IMS is sufficient for its release to the cytosol and activation of the ISR. a** Schematic depicting the localization of the proteases employed in this figure relative to the DELE1 protein during its import into mitochondria. **b** HeLa OMA1 knockout (*ΔOMA1*) cells were transiently transfected with DELE1(3CS) and empty vector control or the indicated 3C protease fused to TOM20, TIM50 or the sorting signal of Su9. Cleavage of the DELE1 protein was analyzed by immunoblotting. **c, d** HeLa cells were transiently transfected with DELE1(3CS) as in (**b**). Localization of the DELE1 protein was analyzed by confocal microscopy after a 2 h treatment with DMSO or CCCP (**c**) or in the context of the co-transfected 3C proteases (**d**). Scale bars, 10 μm. Nuclei (DAPI, blue), mitochondria (MitoTrackerRed, pink), DELE1 (HA, green). **e** The induction of the ISR marker CHOP was measured in HAP1 CHOP[Neon] OMA1 knockout cells by flow cytometry upon transient transfection of the indicated constructs together with mCherry. Relative CHOP[Neon] fluorescence to empty vector transfected cells is shown. Graph depicts mean ± s.d. of $n = 5$ independent experiments. DELE1(ΔMTS) was used as positive control. Statistical significance within DELE1 constructs compared to the respective vector control was assessed by ordinary one-way ANOVA and Dunnett's multiple comparisons correction. DELE1: ns ≥ 0.4019; DELE1(3CS): ****$P < 0.0001$, ns = 0.8269. **f** HeLa *DELE1*[3CS-HA] cells were transfected with the indicated Flag-tagged 3C protease (yellow dots) and cleavage and localization of DELE1 species in mitochondria or the cytosol was analyzed upon subcellular fractionation by immunoblotting.

whereas Su9-3CP would gain access only once DELE1 has reached the mitochondrial matrix (Fig. 4a). DELE1(3CS) was normally processed into M-DELE1 in the steady-state or S-DELE1 upon activation of OMA1 (Supplementary Fig. 4a), and indeed was also cleaved by TOM20-, TIM50- and Su9-3CP (Fig. 4b). Moreover, it localized to mitochondria and was released upon stimulation with CCCP (Fig. 4c). Strikingly, when surveyed for subcellular distribution, we found that co-expression of IMS-targeted or OMM-targeted 3CP led to cytosolic accumulation of DELE1, which did not occur when 3CP was targeted to the matrix (Fig. 4d). Moreover, artificially cleaved and released DELE1(3CS) resulted in an induction of CHOP that was comparable with that observed with a constitutively cytoplasmic version of DELE1 lacking the MTS (Fig. 4e). No activation of CHOP was observed with the corresponding catalytic dead versions[58] of the 3CP constructs or when DELE1 lacking the 3CS was used (Supplementary Fig. 4b). In line with this observation, the cytosolic species of endogenous DELE1[3CS] consisted of the cleavage product rather than immature precursor, as we determined by cellular fractionation using cells in which the endogenous gene was engineered to contain a 3CS (Fig. 4f). In summary, these data demonstrate that cleavage of DELE1 downstream of its extended MTS is sufficient for its mitochondrial release. Moreover, they reinforce the notion that cleavage-triggered release of DELE1 is possible from early mitochondrial destinations (OMM and IMS) but can no longer be accommodated once this section of the protein has reached the matrix.

**DELE1 tracks different forms of mitochondrial import stress.** Given that the requirement of OMA1 for the mitochondrial

release of DELE1 could be bypassed when DELE1 was cleaved by an exogenous protease, we concluded that mitochondrial import and release of DELE1 may be governed by elements in its own amino acid sequence. As DELE1 processing by OMA1 results in the removal of its N-terminal 142 amino acids[12], we suspected that elements in that sequence interfere with release in the absence of OMA1. To test this hypothesis, we generated deletion mutants in this portion of the protein, while sparing the identified presequence required for mitochondrial localization (Fig. 5a). These mutants gave rise to the M-DELE1 species, indicative of MPP processing, and could still be converted into S-DELE1 in an OMA1-dependent manner when mitochondria were stressed (Fig. 5b). Moreover, they normally localized to mitochondria in the steady-state, but compared to wild-type DELE1, these mutants were much more readily released from depolarized mitochondria in OMA1-deficient cells (Fig. 5c). Analysis of the observed mitochondrial release by cellular fractionation revealed that deletion of the presequence-adjacent extended MTS region of DELE1 (37–106), the section between the MTS and the OMA1 cleavage site (108–136), or their combined removal each resulted in an appreciable release from depolarized mitochondria (Supplementary Fig. 5a). Testing the effects of these portions of DELE1 on OMA1-mediated cleavage in the context of polarized mitochondria with inhibited ATP synthase revealed that the extended MTS (37-106) was critical for S-DELE1 generation in this setting (Fig. 5d). The presence of this section of DELE1 even enabled OM-stimulated cleavage of variants in which the presequence was swapped for the effective matrix-targeting presequence of Su9, whereas the corresponding chimera in which the entire DELE1 MTS was exchanged could not be cleaved (Fig. 5e;

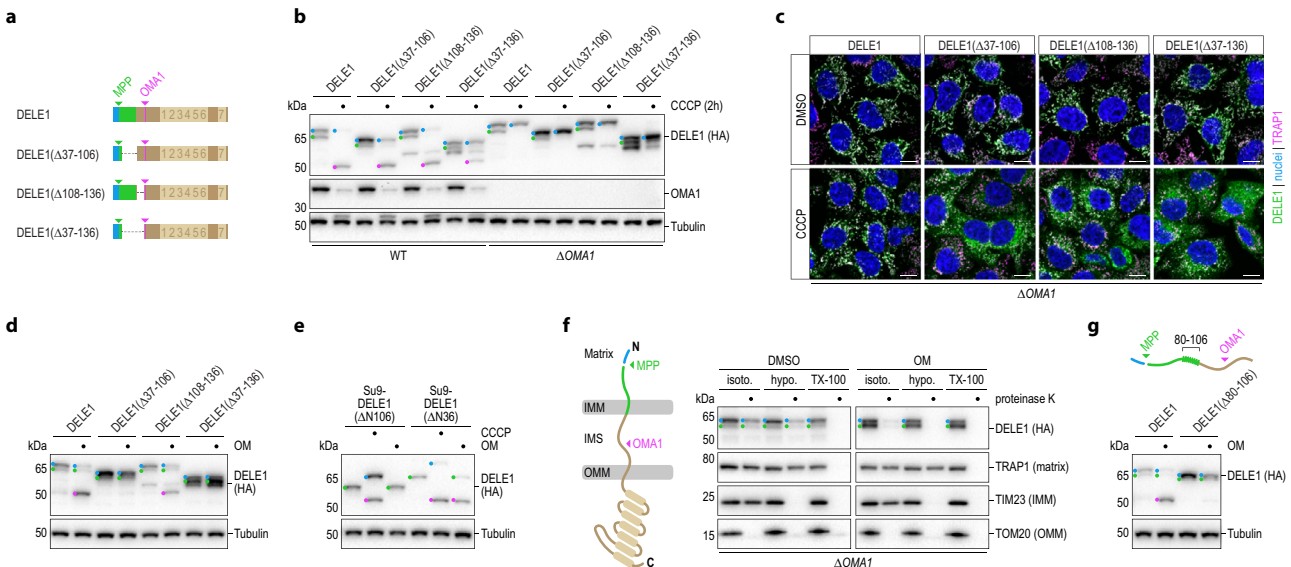

**Fig. 5 Sorting of DELE1 in the context of mitochondrial stress is instructed by cis elements. a** Schematic illustrating DELE1 and the deletions made in its MTS (mitochondria targeting sequence). **b** HeLa WT and *ΔOMA1* cells were transiently transfected with the MTS variants illustrated in (**a**), treated with CCCP as indicated and processing of DELE1 was analyzed by immunoblotting. **c** HeLa *ΔOMA1* cells stably expressing the MTS variants illustrated in (a) were treated with CCCP for 2 h and localization of DELE1 was analyzed by confocal microscopy. Scale bars, 10 μm. Nuclei (DAPI, blue), mitochondria (TRAP1, pink), DELE1 (HA, green). **d** 293T *ΔDELE1* cells were transiently transfected with the DELE1 variants in (**a**), treated with OM for 4 h, and assayed for DELE1 cleavage by immunoblotting. **e** 293T *ΔDELE1* cells were transfected with chimeric constructs in which either only the DELE1 presequence or the entire DELE1 MTS was replaced by the matrix-targeting signal of Su9 (see Supplementary Fig. 5b). Cells were treated with CCCP or OM for 4 h and assayed by immunoblotting. **f** Mitochondria isolated from 293T *DELE1*HA *ΔOMA1* cells treated with DMSO or OM for 4 h were processed and analyzed as in Fig. 2l. **g** DELE1-deficient 293T cells were transfected with full-length DELE1 or DELE1 lacking the identified hydrophobic sequence, treated with OM for 4 h and analyzed by immunoblotting.

Supplementary Fig. 5b). Similarly, this effect of the extended MTS could also not be overcome by strengthening the natural presequence[59] of DELE1 by an arginine substitution (T24R) (Supplementary Fig. 5c). The apparent central role of the DELE1 MTS for OMA1-mediated cleavage in the context of ATP synthase inhibition by OM prompted us to investigate compartment-specific proteinase K sensitivity of DELE1 in this setting. Strikingly, compared to the steady-state the C-terminus of M-DELE1 (alongside L-DELE1) was highly sensitive to proteinase K even in intact mitochondria of OM-treated cells (Fig. 5f). This indicates that the C-terminus of DELE1 molecules whose presequence had already been removed by MPP was still accessible from the cytosol and thus suggests an arrest in sorting in a configuration that spans both membranes[50,60]. This type of import arrest can be promoted by hydrophobic amino acid stretches downstream of the presequence acting as stop-transfer signals at the TIM23 complex of the IMM[60,61]. Hydrophobicity analysis of the DELE1 MTS identified a sequence of mostly uncharged/apolar amino acids between residues 80 and 106, which are predicted to adopt an α-helical configuration[62] and may thus serve a similar purpose when matrix import is compromised by ATP synthase malfunction. In support of this notion, we found that deletion of this segment strongly diminished the ability of OMA1 to cleave DELE1 in cells treated with OM (Fig. 5g; Supplementary Fig. 5d).

While the movement of many precursor proteins across the OMM does not require matrix ATP, it can be critical to drive the translocation of partially folded protein segments[63]. The heme-binding (HB) domain of yeast cytochrome $b_2$ (CYB2) hinders passage through the mitochondrial import machinery due to its stable fold that complicates passage through the TOM complex[64]. We thus wondered whether we could modulate DELE1 import in a similar fashion and thus possibly the window for its cleavage by OMA1. To address this question, we fused the CYB2 HB domain to the C-terminus of DELE1 and monitored its processing into

S-DELE1 when OMA1 was activated by OM. We observed that the addition of an intact HB domain promoted the generation of S-DELE1, whereas DELE1 carrying a destabilized HB mutant (HB*)[64] was processed indistinguishably from wild-type DELE1 (Fig. 6a; Supplementary Fig. 6a). Of note, expression of DELE1-HB in 293T cells already yielded an appreciable level of S-DELE1 formation in the absence of OM. This suggests that partial obstruction of the mitochondrial import pore by challenging proteins might itself trigger the OMA1-DELE1-HRI pathway. However, in this case, cleavage of DELE1-HB was caused by DELE1-HB itself. To test whether DELE1 more generally is able to respond to mitochondrial import problems elicited by challenging mitochondrial precursor proteins, we made use of the observation that overexpression of certain IMM proteins triggers a stress response termed mitoCPR in yeast[8]. To this end, we exogenously expressed mCherry fused to the MTS of mitoCPR-inducing COX5A[8] or COX8A as control and monitored activation of CHOP. Strikingly, while COX8A-mCherry did not result in appreciable CHOP induction regardless of DELE1 status, COX5A-mCherry triggered CHOP more strongly in wild-type cells than in DELE1 knockout cells or cells lacking HRI (Fig. 6b). This phenotype was corrected by the complementation of DELE1-deficient cells with DELE1.

The cleavage and import behavior of DELE1 seems reminiscent of another mitochondrial stress sensor regulated by its sub-mitochondrial localization—PINK1—albeit in the opposite manner: While similarly to DELE1, PINK1 is also cleaved by MPP[65] and a second protease (in this case PARL), its sorting into polarized versus depolarized mitochondria is inverted. In the absence of mitochondrial stress, PINK1 is cleaved by PARL, which results in its mitochondrial release and cytosolic degradation; whereas when mitochondria are depolarized, PINK1 accumulates on the OMM where it initiates mitophagy[66–68]. We thus wondered whether this partitioning could be reversed in

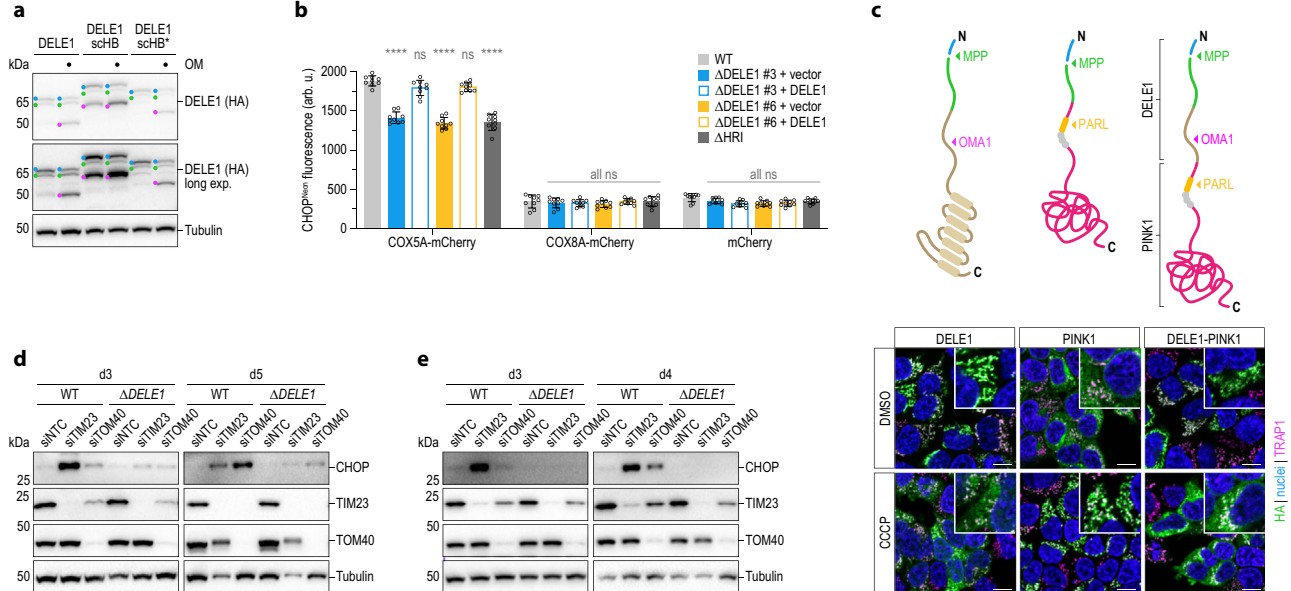

**Fig. 6 DELE1 responds to various types of perturbed mitochondrial protein import. a** 293T cells were transiently transfected with the specified constructs and DELE1 processing upon 4 h OM treatment was analyzed by immunoblotting; scHB, *Saccharomyces cerevisiae* heme-binding domain. scHB*, destabilized HB domain. **b** CHOP$^{Neon}$ fluorescence in the indicated cell lines was analyzed 20 h after transfection of mCherry fused to the sorting signals of COX5A or COX8A. Graph depicts mean ± s.d. of $n = 3$ independent experiments each containing three biological replicates. Statistical significance compared to WT within each construct is shown and was assessed using two-way ANOVA with Tukey's multiple comparisons correction. COX5A-mCherry: ****$P < 0.0001$, ns ≥ 0.1712; COX8A-mCherry: ns ≥ 0.9994; mCherry: ns ≥ 0.3946. **c** 293T cells stably expressing the indicated DELE1-PINK1 chimeras were treated with DMSO or CCCP for 4 h and localization of the indicated HA-tagged proteins was analyzed by confocal microscopy. Scale bars, 10 μm. Nuclei (DAPI, blue), mitochondria (TRAP1, pink), HA (green). **d, e** CHOP induction upon knockdown of TIM23 or TOM40 by siRNA transfection into HeLa (**d**) or 293T (**e**) WT or DELE1 knockout (*ΔDELE1*) cells was analyzed by immunoblotting at the indicated times post-transfection.

a chimeric PINK1 protein by transplantation of the DELE1 N-terminus. In contrast to DELE1, wild-type PINK1 was detected diffusely in the cytosol in the absence of stress, but showed mitochondrial localization when the organelle was perturbed with CCCP (Fig. 6c; Supplementary Fig. 6b). The behavior of the chimeric PINK1 protein carrying the DELE1 N-terminus, however, was reversed: It localized to mitochondria in unstressed cells but could be found in the cytosol when mitochondria were depolarized. As predicted by our prior observations with the DELE1 N-terminus, this cytosolic localization was enabled by OMA1-mediated cleavage in the transplanted sequence (Supplementary Fig. 6c).

Given our observations that DELE1 is highly sensitive to alterations in its own sorting and processing, as well as to the effects of other problematic precursor proteins, we next sought to investigate whether DELE1 can also transmit defects in the mitochondrial import machinery itself. The TOM complex of the OMM represents the initial entry gate for most mitochondrial proteins, while the TIM23 complex of the IMM is required for the sorting of nuclear-encoded matrix proteins, as well as some proteins of the IMM and IMS[4]. Perturbation of respective pore-forming components TIM23 or TOM40 by RNAi resulted in an activation of CHOP mediated by DELE1, as it was blunted in DELE1 knockout cells (Fig. 6d, e). Importantly, this phenotype was not caused by a general inability of the DELE1-deficient cells to induce CHOP downstream of eIF2α phosphorylation, as perturbation of the ER with tunicamycin resulted in a productive generation of CHOP indistinguishable from wild-type cells (Supplementary Fig. 6d). Inhibition of the TOM complex would be predicted to promote the accumulation of immature L-DELE1 precursor protein in the cytosol. In line with this notion, we observed that OMA1 was largely dispensable for CHOP activation in TOM40-depleted cells (Supplementary Fig. 6e). Would cytosolic full-length DELE1 thus be able to bind and stimulate HRI? To

address this notion, we devised a strategy for selective co-immunoprecipitation in the cytosolic fraction through digitonin lysis: this setting allows for the specific analysis of protein-protein interactions that occur in the cytosol through the elimination of mitochondria and other organelles (Supplementary Fig. 6f). We reasoned that moving the C-terminal HA-tag of DELE1 to its N-terminus would hinder the recognition of the MTS by the mitochondrial import machinery and enforce cytosolic expression of the full-length protein, which we indeed observed (Supplementary Fig. 6g). We then expressed this version of DELE1, the C-terminally tagged construct, or a mutant that constitutively localized to the cytosol due to removal of the MTS (DELE1(ΔMTS)) and compared their ability to bind to HRI in the absence of mitochondrial stress. This revealed that cytosolic full-length HA-DELE1 was able to bind to HRI similarly to DELE1(ΔMTS) (Supplementary Fig. 6h). Moreover, it was able to trigger CHOP expression in an HRI-dependent manner (Supplementary Fig. 6i). Together, this indicates that the presence of the MTS in DELE1 neither prevents binding to HRI nor activation of the kinase. In line with these findings, in the setting of TOM40 perturbation, we also observed an association between HRI and endogenous L-DELE1 in the cytosol (Supplementary Fig. 6j, k).

Together, these data establish DELE1 as a critical relay of perturbed mitochondrial import in the human system. We observed that DELE1 can detect multiple types of import defects and signal those to HRI and the ISR in different modes, depending on the site and nature of the respective perturbation.

**DELE1 modulates cell fate in the context of hereditary mito-chondrial defects.** Finally, we sought to study the role of the DELE1 pathway in the context of mitochondrial import and precursor maturation defects observed in hereditary neurode-generative disorders. Our genome-wide phenotypic screen

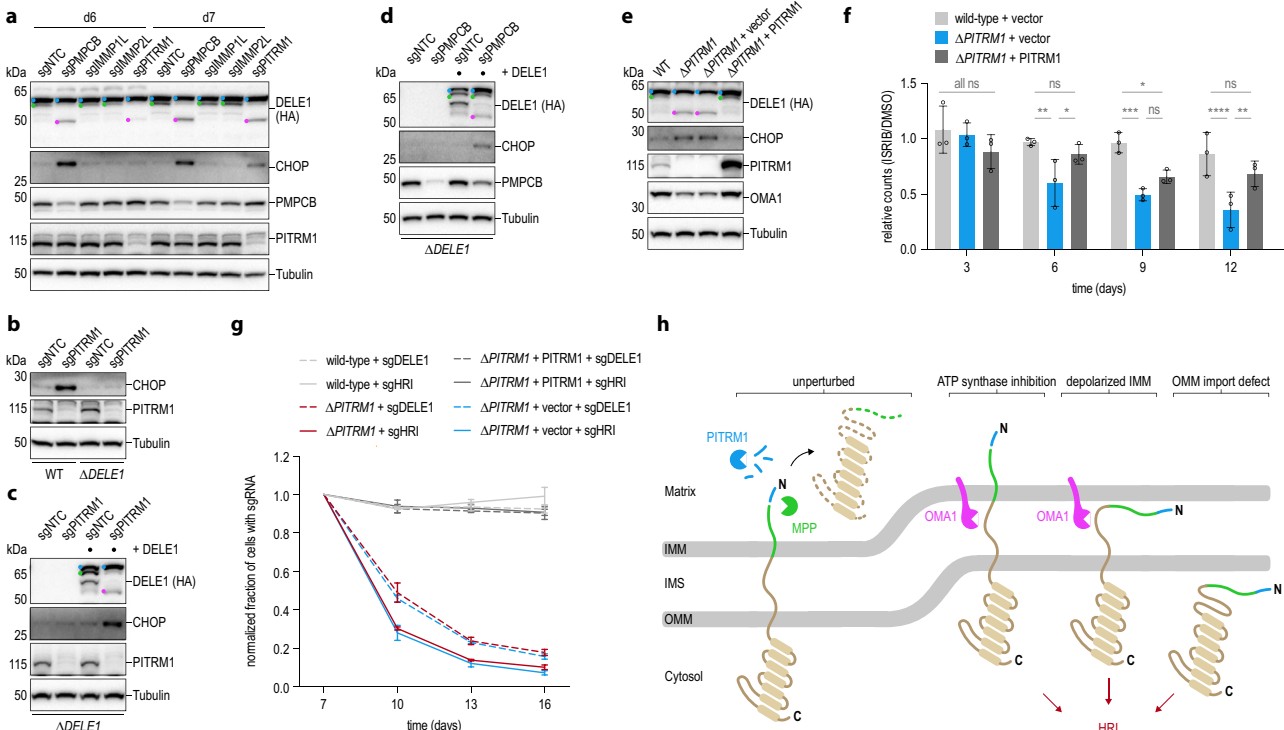

**Fig. 7 Mitochondrial stress induced by deficiencies in MPP or PITRM1 imposes a cytoprotective state of ISR signaling via DELE1. a** HeLa *DELE1*[HA] cells exposed to the specified sgRNAs or a non-targeting control (NTC). DELE1 processing and CHOP induction analyzed by immunoblotting as indicated. **b** HeLa WT and *ΔDELE1* cells were transiently exposed to an sgRNA directed against PITRM1 (or NTC sgRNA) and induction of CHOP was analyzed by immunoblotting seven days post-transfection. **c, d** HAP1 DELE1 knockout (*ΔDELE1*) or stably reconstituted cells were infected with lentivirus encoding a NTC sgRNA or sgRNAs targeting PITRM1 (**c**) or PMPCB (**d**). CHOP levels were analyzed by immunoblotting seven (**c**) or six (**d**) days post transduction. (**e**) HeLa *DELE1*[HA] WT, clonal PITRM1 knockout cells (*ΔPITRM1*) and clonal PITRM1 knockout cells reconstituted with PITRM1 or empty vector (*ΔPITRM1* + PITRM1/ vector) were analyzed for DELE1 cleavage and CHOP levels by immunoblotting. **f** HeLa *DELE1*[HA] cells of the indicated genotypes were cultured in medium containing ISRIB or DMSO as control and cell growth was followed over time. Cells were passaged and counted every 3 days. Mean ± s.d. of n = 3 independent biological replicates (separately seeded and cultured wells for each genotype and treatment) is shown. Statistical significance was assessed using two-way ANOVA with Tukey's multiple comparisons correction. Day 3: ns ≥ 0.1336; Day 6: **P = 0.0027, *P = 0.0396, ns = 0.5268; Day 9: ***P = 0.0002, *P = 0.0121, ns = 0.2491; Day 12: ****P < 0.0001, **P = 0.0081, ns = 0.1951. **g** HeLa *DELE1*[HA] cells of the indicated genotypes were infected with lentivirus encoding sgRNAs directed against the specified genes or a non-targeting control together with tRFP. The fraction of tRFP-positive cells was followed over time. The percentage of sgDELE1 or sgHRI-containing cells was normalized to the non-targeting control and to day 7 for each cell line. Mean ± s.d. of n = 3 independent biological replicates (separately cultured wells for each sgRNA and genotype) is shown. **h** Model of the fates of DELE1 in the steady-state and in the context of different types of perturbed mitochondrial protein import and processing.

predicted that DELE1 signaling might be affected by the activity of PITRM1 (Fig. 1f). Deleterious mutations in *PITRM1* have been identified in multiple families suffering from a progressive neurodegenerative phenotype with neurological manifestations[27–29]. It has been proposed that loss of PITRM1 installs a cytoprotective response[69] but whether this involves signaling through DELE1 remains unexplored. Given that PITRM1 degrades presequences released by MPP, accumulation of which can cause a backlog that impairs MPP activity[16,26], we reasoned that inactivating mutations in MPP, too, may affect DELE1. MPP is highly essential, but hypomorphic alleles have been reported, which lead to neurodegeneration in early childhood[70]. To test these hypotheses, we exposed cells expressing endogenously tagged DELE1 to sgRNAs targeting *PITRM1* or *PMPCB* (encoding the catalytic subunit of MPP) and monitored DELE1 cleavage. Strikingly, sgPMPCB-treated and sgPITRM1-treated cells showed processing of DELE1 into S-DELE1 (Fig. 7a). The observed generation of S-DELE1 also coincided with the induction of CHOP in these cells, indicative of ISR activation. Induction of CHOP following PITRM1 depletion was blunted in DELE1-deficient cells (Fig. 7b). This effect was reverted when the cells were complemented with a DELE1 cDNA (Fig. 7c), indicating that the response to PITRM1 defects involves

DELE1 signaling. We observed similar effects for PMPCB, although the sustained knockout efficiency was lower for this enzyme (Fig. 7d). Interestingly and reminiscent of the effects observed with TOM40 inhibition, DELE1 signaling elicited by PITRM1 defects was not dependent on OMA1, as L-DELE1 could still associate with HRI in the absence of the protease (Supplementary Fig. 7a, b).

These findings prompted us to investigate whether the DELE1 signaling axis is also involved in the cellular response to chronic deficiencies in PITRM1, as experienced in the patient setting[69]. Indeed, clonal PITRM1 knockout cells showed a persistent low level of DELE1 cleavage and CHOP activation that was rescued by complementation with a PITRM1 cDNA (Fig. 7e). Next, we sought to test whether this might reflect a cytoprotective adaptation to the stress imposed by PITRM1 deficiency. As DELE1 relays mitochondrial stress to CHOP via HRI-dependent activation of the ISR[11], we monitored the growth of PITRM1-proficient and PITRM1-deficient cells in the presence of the ISR inhibitor ISRIB[71]. This showed that PITRM1-defective cells indeed had a fitness disadvantage in the presence of the ISR inhibitor (Fig. 7f; Supplementary Fig. 7c). To determine the role of DELE1 signaling in this context, we depleted DELE1 or HRI

via sgRNAs from wild-type cells, PITRM1 knockouts and reconstituted cells and monitored their fitness in a competitive growth assay by flow cytometry. This showed that cells which could not engage ISR signaling in conjunction with PITRM1 deficiency due to the presence of sgRNAs targeting DELE1 or HRI were depleted from the culture over time (Fig. 7g; Supplementary Fig. 7d). In summary, these data indicate that DELE1 is able to track perturbations in certain proteolytic activities within mitochondria dysregulated in human disease and that it can promote a cytoprotective state of ISR activation.

## Discussion

It has long been appreciated that mitochondria are particularly vulnerable cellular structures that need to be protected from a wide range of potential external and internal insults. These include but are not limited to perturbed oxidative phosphorylation and production of reactive oxygen species, nutrient deprivation, hypoxia, mitochondrial depolarization, defective protein import or mitochondrial protein synthesis, unfolded, damaged and aggregation-prone proteins, mtDNA defects and mitonuclear imbalances, as well as microbial pathogens[10,13,72,73]. To rationalize how such a diverse set of perturbations could possibly be sensed and responded to, it was proposed that most, if not all, of these stresses ultimately impinge on the conserved process of mitochondrial protein import[2,55,72,74]. In nematodes, this is elegantly exploited by the dual-targeted transcription factor ATFS-1, which is imported into unperturbed mitochondria but relocalizes to the nucleus when mitochondrial protein import is impaired[5]. In a similarly subtle manner, the mitophagy and Parkinson's disease kinase PINK1 is regulated by differential cleavage reactions during its import into healthy versus perturbed mitochondria[50,51,75]. Here we show that the factor that signals a broad range of mitochondrial insults in the human system – DELE1 – combines elements of both mechanisms to alert the cell about impending harm. Our data indicate that the cell ensures a continuous flux of DELE1 precursors through the mitochondrial import and processing machineries by perpetual de novo synthesis and breakdown of the protein. While this may appear like a wasteful strategy, coupling of the DELE1 stress signaling function to its own trafficking and maturation provides a clever solution to keep a tab on mitochondrial functionality (Fig. 7h).

We demonstrate that in the absence of mitochondrial stress, DELE1 follows the presequence pathway into the mitochondrial matrix, where presequences are removed by MPP and subsequently degraded by PITRM1[31]. For DELE1, cleavage occurs after a short N-terminal fragment, removal of which gives rise to M-DELE1. It is currently unclear which functions inside the mitochondrial matrix might be accommodated by M-DELE1. Although ATFS-1 is degraded by the matrix-resident LON protease upon mitochondrial import[5], it was later found to be able to also associate with the mitochondrial genome and negatively regulate the abundance of mtDNA-encoded transcripts of oxidative phosphorylation genes[76]. Moreover, in the context of stress, multiple isoforms and/or cleavage fragments of ATFS-1 become enriched in mitochondria. The genetic screen for DELE1 regulators identified a role for CLPP (Fig. 1f; Supplementary Fig. 7e), a matrix-resident protease involved in mitochondrial ribosome assembly and the degradation of nonassembled or misfolded proteins[23]. Interestingly, we also observed that besides M-DELE1, additional species of DELE1 can be generated inside mitochondria (Fig. 3g), which we did not observe for cytosolic HA-DELE1 (Supplementary Fig. 6h). As this is the case for endogenous DELE1 as well as DELE1 expressed from a cDNA, these likely result from additional proteolytic events in the IMS and/or matrix. Going forward, it will be of interest to explore if

DELE1 serves functions beyond its role as a stress relay upon import into mitochondria and how these might be connected to possible additional processing steps.

Presequences are usually in the range of 20 to 40 amino acids in length[77]. In line with this, our data indicate that the N-terminal 25-36 amino acids of the DELE1 MTS already promote mitochondrial localization (Supplementary Fig. 2c; Fig. 5c). This raises the question why DELE1 has evolved to contain such an unusually long MTS of 100-115 amino acids. We propose that the MTS of DELE1 accommodates two distinct functions: The N-terminal presequence is required to direct mitochondrial import and sorting of DELE1 whereas the downstream portion of the MTS counteracts release of DELE1 from stressed mitochondria in the absence of OMA1 activity (Fig. 5). It has been reported that the ability of mitochondrial precursor proteins to be imported into partially depolarized mitochondria depends on the net charge of the MTS[35], which is a mechanism by which ATFS-1 senses mitochondrial insults[55]. However, the mature portion of matrix precursors can also determine sensitivity towards a decline in the IMM potential[78]. Cis elements like stably folding domains and sequence motifs that hinder transport across the mitochondrial membranes gain particular importance in settings like perturbation of the ATP synthase, when matrix ATP needed for the complete import of precursors may become limiting[38,63,79]. Cleavage into S-DELE1 in the context of OM treatment was promoted by the addition of a tightly folded HB domain to the DELE1 C-terminus, presumably stalling translocation across the mitochondrial membranes[64] (Fig. 6a). Regarding unmodified DELE1, we observed that when the ATP synthase was perturbed, some DELE1 molecules that had undergone processing by MPP appeared to have become arrested in the import process with the C-terminus facing the cytosol. This configuration has also been observed for other mitochondrial proteins in the context of perturbed matrix import[38,63]. Our data reveal that in this setting, DELE1 cleavage by OMA1 strongly relied on a short hydrophobic region with predicted α-helical configuration near the end of the extended DELE1 MTS, which may thus act akin to a stop-transfer signal (Fig. 5f, g). These modes of DELE1 regulation are reminiscent of the sorting and protease processing of PINK1, whose activity, too, is controlled by mitochondrial import fidelity, nurturing the possibility of coordination between these two important mitochondrial stress relays[10,50,60,75].

Overexpression of COX5A impedes TIM23-dependent precursor import and maturation in yeast[8]. Interestingly, its expression in human cells triggered the induction of the ISR marker CHOP in a DELE1-dependent manner (Fig. 6b). However, the absence of DELE1 did not completely abrogate this cellular response, hinting at the existence of additional unidentified import stress sensing modules in the human system. This will be an exciting avenue going forward, as several quality control mechanisms acting at the level of the mitochondrial import pore have been discovered in yeast[6–9,80], for some of which obvious orthologous genes in mammals are either lacking or activity in the human system remains to be demonstrated[10].

Our data indicate that uncleaved full-length DELE1 is able to bind and activate HRI to a similar extent as S-DELE1 and that this can signal mitochondrial defects at the level of the TOM complex (Supplementary Fig. 6). Prolonged depolarization of mitochondria also results in the accumulation L-DELE1 (Fig. 2a), some of which would be predicted not to enter into mitochondria. This might be similar to human disease settings with imbalanced expression of oxidative phosphorylation components[76]. It is tempting to speculate that the role of OMA1-mediated severing of the DELE1 N-terminus might serve as an additional safeguard against unwanted ISR activation in the absence of an appropriate stress signal for OMA1, or tune the degree of the cellular response. That mitochondrial stress

signaling can be maladaptive rather than beneficial in a context-dependent manner has been documented[1,10–12,81]. It will be of interest to identify the precise mode of OMA1 activation and under which circumstances mitochondrial import stress and OMA1 activation might be uncoupled, also in light of OMA1 being pursued as a drug target[82,83].

Beyond mitochondrial import defects, we find that DELE1 can also act on perturbations in proteases that function in the maturation of mitochondrial matrix precursors—DELE1 itself being one of their clients. While *PMPCB* defects result in severe phenotypes and early childhood lethality[70], mutations in *PITRM1* yield a milder state of mitochondrial perturbation, in which DELE1 participates in the installation of a cytoprotective activation of the ISR. As for MPP[84], loss of PITRM1 has also been reported to upregulate the expression of mitochondrial chaperones and proteases[69] and it will be important to identify the involved pathways. Interestingly, besides mitochondrial presequences, PITRM1 also degrades Aβ peptides, which are associated with Alzheimer's disease pathology and can accumulate in mitochondria, where they perturb mitochondrial function and induce neuronal death[85]. Strikingly, it has been found that enhanced mitochondrial proteostasis through mitochondrial stress signaling protects against Aβ toxicity[74]. In light of the identified connections, modulation of the OMA1-DELE1-HRI pathway might be beneficial in certain human disease settings, including Alzheimer's pathology and PITRM1/PMPCB-related disorders.

Important work in *C. elegans*[5] and yeast[6–9] has revolutionized our understanding of how mito-nuclear communication preserves mitochondrial fidelity in time of stress. Here we add to these insights by describing a mechanism in the human system that couples the sensing and signaling of mitochondrial dysfunction to the ancient conserved processes of protein import and processing in these organelles.

## Methods

**Cell lines.** The HAP1 cell line was generated in the Brummelkamp laboratory[86] and was cultured in IMDM (Iscove's modified Dulbecco's medium; Thermo Fisher Scientific) containing 10% fetal calf serum (FCS; heat inactivated) and penicillin-streptomycin-glutamine. HeLa (ATCC CCL-2) and HEK293T (293T, ATCC CRL-3216) cell lines were cultured in DMEM (Dulbecco's modified Eagle's medium; Thermo Fisher Scientific) supplemented with 10% fetal calf serum (FCS; heat inactivated) and penicillin-streptomycin-glutamine. Cell lines were kind gifts from T. Brummelkamp.

**Generation of genome-edited cell lines.** The CRISPR-Cas9 system was used to insert tags into endogenous loci and to generate knockout cell lines.

293T *DELE1*[HA] cells were described previously[11] and HeLa *DELE1*[HA], HAP1 *DELE1*[HA], HAP1 *DELE1*[ALFA] (donor containing a glycine serine linker, the sequence for the HRV 3C site and a triple ALFA tag[19]) and HeLa *DELE1*[HiBiT] (donor containing a glycine serine linker and a single HiBiT sequence (Promega)) cells were generated accordingly.

HeLa *DELE1*[HA] cells were used to generate HeLa *DELE1*[3CS-HA] cells analogously using a pX330 vector (Addgene, #42230) encoding for an sgRNA directed at exon 6 of the DELE1 gene. The donor plasmid contained homology arms of approximately 840 bp upstream and 440 bp downstream of the sgRNA recognition site, surrounding the sequence of the HRV 3 C recognition site. The homology arms were flanked by an sgRNA sequence of the *tia1l* locus from *Danio rerio* and the donor cassette was released within the cell by co-transfection of a pX330 plasmid containing an sgRNA targeting the *tia1l* site[43].

Polyclonal knock-in cell lines for other HiBiT®-tagged proteins were generated in HeLa cells by transient transfection of a donor vector containing a glycine-serine linker and the HiBiT® sequence (Promega) followed by a polyA signal, pause site, human phosphoglycerate kinase (hPGK) promoter, puromycin N-acetyltransferase and polyA signal. The donor cassette was flanked by synthetic sgRNA recognition sites[87] and released by co-transfection of a pX330 plasmid encoding a corresponding sgRNA. Cas9-catalyzed DNA cleavage in the gene of interest was mediated by co-transfection of a pX330 plasmid containing an sgRNA against the C-terminus of the open reading frame of the targeted gene. Cells were selected with puromycin to enrich for knock-in events.

Knockout cell lines were generated by transient transfection of pX330 plasmids containing the sgRNA of interest. For a list of oligonucleotides see Supplementary Table 1.

Co-transfection of puromycin or blasticidin resistance-encoding plasmids allowed the selection of transfected cells using puromycin or blasticidin (Invivogen).

Alternatively, cell lines were transduced with lentivirus produced in 293T cells by transfection of plasmids pL-CRISPR.EFS.tRFP (Addgene, #57819) or lentiCRISPR v2 (Addgene, #52961) together with lentiviral packaging plasmids (pMDLg/pRRE, pRSV-Rev and VSV.G). Cells were transduced in the presence of protamine sulfate following standard procedures.

Polyclonal populations were used for phenotypic assays and where indicated, single cell clones were generated and analyzed by PCR, Sanger sequencing and/or immunoblotting.

**Haploid genetic screen.** Regulators of DELE1 levels were identified through haploid genetic screening as previously described[11]. Ultra-deep genome-wide mutagenesis of haploid HAP1 *DELE1*[ALFA] cells was performed using a variant of the gene-trap retrovirus[88] containing BFP, produced in 293T cells and concentrated by ultracentrifugation (2 h at 88,800 × *g* at 4 °C). After three consecutive transductions, the library of mutant cells was expanded to 20 T175 flasks, harvested by trypsinization (Trypsin-EDTA 0.25%; Gibco) and fixed in BD fix buffer I (BD Biosciences) for 10 min at 37 °C. Permeabilization of approximately 2 × 10⁹ cells was carried out using BD perm buffer III (BD Biosciences) for 30 min on ice, followed by staining of the DELE1[ALFA] protein with the FluoTag®-X2 anti-ALFA Atto488 (NanoTag N1502-At488-500µL; for a list of all antibodies used in this study see Supplementary Table 2) antibody at a dilution of 1:500 in PBS + 1% FCS + 1% BSA for 2.5 h at RT, rotating. Included in the primary antibody dilution was DAPI (Sigma-Aldrich, D9542) at a final concentration of 2.5 µg/mL to allow gating for haploid cells based on DNA content[89]. An unstained control as well as unedited wild-type HAP1 cells were used to assess staining specificity. Washing steps were performed using PBS + 1% FCS. 1.19 × 10⁷ DELE1[ALFA]-low and 1.21 × 10⁷ DELE1[ALFA]-high cells were sorted on a BD Fusion cell sorter (BD Biosciences).

The sorted cell populations were used for genomic DNA extraction (QIAmp DNA Mini Kit, Qiagen, 51306) following the manufacturer's instruction except that de-crosslinking was performed at 56 °C overnight. Gene-trap insertion sites were cloned as described previously[11] and sequenced with a read length of 50 bp on a HiSeq1500 (Illumina) using single-end mode. Demultiplexed sequencing reads were subsequently aligned to the human genome (hg19) and tallied as described[18]. Briefly, reads were first aligned to the genome (allowing one mismatch) using Bowtie[90]. Multiple reads mapping to identical coordinates were counted only once. For enrichment of insertions in RefSeq genes using intersectBED[91] only insertions occurring in the sense direction of the gene were considered. Significant regulators of DELE1 were identified by comparing the number of unique sense integrations mapping to a query gene to the number of all other mutations in that population between the DELE1[ALFA]-high and DELE1[ALFA]-low cells using a two-sided Fisher's exact test. Multiple testing was accounted for using Benjamini–Hochberg false discovery rate correction. For every gene, the fraction of unique insertions mapping to that gene identified in the DELE1[ALFA]-high population was divided by the corresponding fraction of unique insertions in the DELE[ALFA]-low population. For visualization purposes, per gene, this mutation ratio (*y*-axis) was then plotted against the combined number of unique mutations identified in that gene in the DELE1[ALFA]-high and DELE1[ALFA]-low populations (*x*-axis). The resulting scatter plots were generated using GraphPad Prism.

**Cloning.** Universal human reference RNA (Thermo Fisher Scientific, QS0639) was used to generate cDNA with the Thermo Scientific™ RevertAid First Strand cDNA Synthesis Kit (Thermo Fisher Scientific), from which coding sequences of genes and sorting signals of OPA1(1-209) and MICU1(1-60)[53] were amplified by PCR. DELE1 and DELE1(ΔMTS) have been described before[11] and variants of DELE1 were generated from the DELE1 coding sequence by PCR. The following sequences were derived from plasmids obtained from Addgene: Su9(1-69) (#23214), TEV protease (#116062), PINK1 (#13320). Sequences for CLUH, COX8A(1-36), 3×ALFA-tag, split mNG2, scHB, scCOX5A(1-118), scPSD1(1-108) and 3C protease were codon-optimized and obtained by gene synthesis (IDT; Thermo Fisher Scientific). Restriction digest and ligation into destination vectors were performed following standard procedures. Cloned constructs were verified by Sanger sequencing.

**Generation of stable cell lines.** For stable expression of coding sequences, 293T cells were transfected with retroviral vectors, pAdvantage (Clontech) and the packaging plasmids pCMV-VSV.G and pGAG-POL.

For knockdown of TOM40 by shRNAs[92], 293T cells were transfected with the lentiviral pLKO vector (Addgene, #21915) together with lentiviral packaging plasmids (pMDLg/pRRE, pRSV-Rev and VSV.G).

Viral supernatant was collected 48 h after transfection. Cell lines were transduced with filtered viral supernatant in the presence of protamine sulfate and selected 24 h after transduction with puromycin or blasticidin (Invivogen).

**Treatments and transfections.** Cells were treated for the indicated times with 20 µM CCCP (carbonyl cyanide *m*-chlorophenyl hydrazone; Sigma-Aldrich, C2759), 10 µM OM (oligomycin A; MedChem Express, HY-16589; Sigma-Aldrich, 75351),

20 μg/mL cycloheximide (Sigma-Aldrich, C4859), 10 μM tunicamycin (MedChem Express, HY-A0098), 200 nM ISRIB (Sigma-Aldrich, SML0843).

Cells were plated 24 h before transfection with polyethylenimine (PEI 25000, Polysciences) or turbofectin (OriGene Technologies). Cells were treated, harvested, passaged or placed on selection medium 24 h after transfection.

Transfections of siRNAs (siNTC, D-001810-10-05, Horizon Discovery; siTOMM40, M-012732-00-0010, Horizon Discovery; siTIMM23, 1299001, Thermo Fisher Scientific) were performed using Lipofectamine RNAiMAX Transfection Reagent (Thermo Fisher Scientific) following the manufacturer's instructions.

For induction of shRNAs from the pLKO vector, cells were treated with 500 ng/mL doxycycline hyclate (Biomol, Cay14422-1) for 5 days.

For growth assays, cells were counted and seeded at equal numbers in triplicate wells on day 0 in DMSO (control) or 200 nM ISRIB-containing medium. Cells were passaged every 2–3 days and counted using Countess® Cell Counting Chamber Slides (Thermo Fisher Scientific, C10228). Counts shown in bar graphs represent the ratio of ISRIB-treated cells versus DMSO-treated controls.

**Immunofluorescence.** Cells were plated on ploy-L-lysine-coated coverslips or in chambered slides (ibidi, 80826) 24 h before treatment or transfection. Where indicated, cells were stained with 100 nM MitoTrackerRed (Molecular Probes MitoTracker Red CMXRos, Thermo Fisher Scientific, M7512) for 1 h at 37 °C and washed twice with PBS before treatment or fixation. After a PBS wash, cells were fixed at RT for 10 min with 3.7% paraformaldehyde. PBS containing 0.05% TritonX-100 was used for permeabilization for 30 min at RT and cells were blocked in 10% FCS in PBS for 30 min at RT. Incubation in primary antibody dilution was carried out at RT for a minimum of 1 h followed by a 1 h incubation with a dilution of fluorophore-conjugated secondary antibodies in the dark. Antibodies were diluted in PBS containing 10% FCS and PBS was used for washing steps. In one of the final washing steps, DAPI was included at a final concentration of 1 μg/mL. Roti-Mount FluorCare (Carl Roth, HP19.1) was used to mount coverslips. Images were acquired with a Zeiss Observer.Z1 confocal microscope (Carl Zeiss) using a 63× oil immersion objective and the ZEN2009 software (Carl Zeiss). Live cell imaging was performed on a Leica sp8 using a 63× glycerol immersion objective and LASX software (Leica). Images were analyzed using Fiji[93].

**Immunoblotting.** Cells were washed with PBS and lysed with SDS sample buffer. Lysates were boiled at 99 °C for 10 min. Bolt gradient gels (4–12%) and the Bolt gel electrophoresis system (Thermo Fisher Scientific) were used to separate proteins by gel electrophoresis. Proteins were transferred to PVDF (polyvinylidene fluoride) membranes, which were blocked in 4% milk (Carl Roth, T145.1) in TBST and incubated with the indicated antibodies. HRP-conjugated secondary antibodies (Bio-Rad), the Bio-Rad ChemiDoc™ MP imaging system and Image Lab software were used to detect proteins.

**Immunoprecipitation.** Cells were washed with PBS and lysed with DISC (30 mM Tris-HCl pH 7.5, 150 mM NaCl, 10% glycerol) buffer containing 1% TritonX-100 or 0.02% digitonin (Sigma-Aldrich, D141) freshly supplemented with cOmplete Protease Inhibitor Cocktail (Roche, 11697498001) for 20 min on ice. Lysates were cleared twice by centrifugation at $20,000 \times g$ for 15 min at 4 °C. Prior to and post lysis aliquots of the cell suspension and supernatant, respectively, were collected as input. The anti-EIF2AK1 antibody (ProteinTech, 20499-1-AP) or the anti-mCherry antibody (ProteinTech, 26765-1-AP) diluted in lysis buffer were coupled to Pierce™ Protein A Agarose beads (Thermo Fisher Scientific, 20333) for 1 h at 4 °C on a rotator. After coupling, beads were washed three times with lysis buffer and incubated with cell lysates for 2 h at 4 °C on a rotator. Subsequently, beads were washed three times with DISC containing 1% Triton-X100 before the addition of SDS sample buffer and boiling at 99 °C for 10 min.

**Flow cytometry.** Analysis of DELE1[ALFA] levels was performed by staining the DELE1[ALFA] protein using the FluoTag®-X2 anti-ALFA Atto488 antibody as described above (haploid genetic screen).

Analysis of CHOP[Neon] levels was performed on living cells after detachment using Trypsin-EDTA (0.25%, Gibco). For analysis of CHOP[Neon] levels in transiently transfected cells, a plasmid encoding for mCherry was co-transfected to identify transfected cells and cells were analyzed 20–48 h after transfection. The mean mNeon intensity of transfected cells (top 10% of mCherry-positive cells) was divided by the mean mNeon intensity of untransfected (mCherry-negative) cells. Where absolute values are shown, background fluorescence of unstained or untransfected cells was subtracted.

For the comparative growth analysis Hap1 DELE1[ALFA] or HeLa DELE1[HA] cells of the indicated genotypes were infected at low multiplicity of infection with the construct pL-CRISPR.EFS.tRFP (Addgene, #57819) containing an sgRNA against the indicated genes or a non-targeting control (NTC) sgRNA. The percentage of tRFP-positive, sgRNA-containing cells of three independently cultured wells for each genotype and sgRNA combination was normalized to the NTC sgRNA and day 7 measurements and then followed over time.

Analytical flow cytometry experiments were measured on a BD LSRFortessa (BD Biosciences). The BD FACS DIVA (BD Biosciences) or FlowJo (BD) software was used to analyze data.

**Phylogenetic alignment.** Amino acid alignment of DELE1 sequences (Human, Q14154; Macaque, H9EXR5; Cow, Q0VD37; Mouse, Q9DCV6; Frog, A0A1L8GR41; Salmon, B5X268) was performed using UniProt/clustalo[94].

**Isolation of mitochondria.** For analysis of the mitochondrial and cytosolic proteome by immunoblotting, mitochondria were isolated using the Mitochondria Isolation Kit for Cultured Cells (Thermo Fisher Scientific, 89874) following the manufacturer's instructions. Alternatively, cells were resuspended in cold homogenization buffer (220 mM mannitol, 70 mM sucrose, 20 mM HEPES/KOH pH 7.4, 1 mM EDTA) and thrice passed through a needle with 10 strokes on ice. Mitochondria were separated from the cytosolic fraction by differential centrifugation. Two initial centrifugation steps to pellet nuclei and intact cells were performed at $700 \times g$ for 10 min and 5 min at 4 °C. The supernatant was subsequently centrifuged at $8000 \times g$ for 15 min at 4 °C. The supernatant (i.e. cytosolic fraction) was transferred to a microfuge tube and the mitochondrial pellet was washed with homogenization buffer. After an additional centrifugation step at $8000 \times g$ for 15 min at 4 °C the cytosolic fraction was transferred to a new microfuge tube and SDS sample buffer was added. The mitochondrial pellet was lysed in SDS sample buffer. Immunoblotting was performed as described above.

For proteinase K digestion, mitochondria were isolated using a dounce homogenizer, the above-described homogenization buffer and differential centrifugation. To analyze proteinase K accessibility of mitochondrial proteins, after the second centrifugation step at $8000 \times g$, mitochondria were resuspended in SEM buffer (10 mM MOPS-Tris pH7.2, 250 mM sucrose, 1 mM EDTA), EM buffer (10 mM MOPS-Tris pH 7.2, 1 mM EDTA), or SEM buffer + 0.5% TritonX-100 and incubated on ice for 10 min. Proteinase K (50 μg/mL; Sigma-Aldrich, P6556) was added and digestion was performed for 10 min on ice. To stop the reaction, phenylmethansulfonylfluorid (PMSF, Sigma-Aldrich, P7626) was added to a final concentration of 1 mM. After a 20 min incubation step on ice, SDS sample buffer was added and samples were boiled at 99 °C for 10 min.

**Split luciferase assay.** For the split luciferase assay the NanoLuc® Binary Technology (NanoBiT; Promega) was used. The endogenously tagged HiBiT cell lines were seeded in a 96-well tissue culture plate and transiently transfected 24 h after seeding with the indicated LargeBiT vectors. 24 h after transfection the cells were detached with Trypsin-EDTA (0.25%; Gibco), resuspended in 100 μL OptiMEM (Gibco) + 2% FCS and transferred to a white 96-well microtiter plate (Corning). The Nano-Glo® Live Cell Assay System (Promega, N2011) was used according to the manufacturer's protocol. The luminescence signal was recorded immediately with a TECAN Spark microplate reader using the SparkControl software and a signal accumulation time of 2.5 s.

**Statistics and reproducibility.** Data are shown as mean ± s.d. and statistical parameters are reported in the figure legends. For the haploid genetic screen (Fig. 1f), a total of $n = 2.4 \times 10^7$ single cells were analyzed. A two-sided Fisher's exact test was used to calculate enrichment of mutations in the high or low channel. P-values were FDR-corrected using the Benjamini–Hochberg method as described above (haploid genetic screen). GraphPad Prism 9 software was used to generate graphs and to calculate statistical significance. P-values below 0.05 were considered significant (*$P < 0.05$, **$P < 0.01$, ***$P < 0.001$ and ****$P < 0.0001$). Data are representative of the following numbers of independent experiments: Fig. 1a, $n = 2$; Fig. 1c, $n = 3$; Fig. 1d, $n = 3$; Fig. 1e, $n = 3$; Fig. 1g, $n = 3$; Fig. 1h, $n = 4$; Fig. 1i, $n = 3$; Fig. 1j, $n = 2$; Fig. 1k, $n = 2$; Fig. 2a, $n = 2$; Fig. 2b, $n = 3$; Fig. 2c, $n = 3$; Fig. 2e, $n = 3$; Fig. 2f, $n = 2$; Fig. 2i, $n = 2$; Fig. 2j, $n = 3$; Fig. 2k, $n = 3$; Fig. 2l, $n = 5$; Fig. 3a, $n = 3$; Fig. 3b, $n = 3$; Fig. 3c, $n = 3$; Fig. 3e, $n = 3$; Fig. 3f, $n = 2$; Fig. 3g, $n = 3$; Fig. 4b, $n = 3$; Fig. 4c, d, $n = 3$; Fig. 4e, $n = 3$; Fig. 4f, $n = 4$; Fig. 5b, $n = 2$; Fig. 5c, $n = 2$; Fig. 5d, $n = 3$; Fig. 5e, $n = 3$; Fig. 5f, $n = 3$; Fig. 5g, $n = 3$; Fig. 6a, $n = 3$; Fig. 6b, $n = 3$; Fig. 6c, $n = 2$; Fig. 6d, $n = 4$ (d3), $n = 2$ (d5); Fig. 6e, $n = 3$ (d3), $n = 4$ (d4); Fig. 7a, $n = 3$; Fig. 7b, $n = 2$; Fig. 7c, $n = 4$; Fig. 7d, $n = 4$; Fig. 7e, $n = 3$; Fig. 7f, $n = 2$; Fig. 7g, $n = 4$; Supplementary Fig. 1b, $n = 3$; Supplementary Fig. 1c, $n = 2$; Supplementary Fig. 2a, $n = 3$; Supplementary Fig. 2b, $n = 4$; Supplementary Fig. 2c, $n = 2$; Supplementary Fig. 2d, $n = 5$; Supplementary Fig. 3a, $n = 2$; Supplementary Fig. 3b, $n = 2$; Supplementary Fig. 3c, $n = 3$; Supplementary Fig. 4a, $n = 3$; Supplementary Fig. 4b, $n = 3$; Supplementary Fig. 5a, $n = 2$; Supplementary Fig. 5c, $n = 3$; Supplementary Fig. 5d, $n = 3$; Supplementary Fig. 6a, $n = 2$; Supplementary Fig. 6b, $n = 3$; Supplementary Fig. 6c, $n = 3$; Supplementary Fig. 6d, $n = 2$; Supplementary Fig. 6e, $n = 2$; Supplementary Fig. 6g, $n = 2$; Supplementary Fig. 6h, $n = 2$; Supplementary Fig. 6i, $n = 3$; Supplementary Fig. 6j, $n = 2$; Supplementary Fig. 6k, $n = 2$; Supplementary Fig. 7a, $n = 4$; Supplementary Fig. 7b, $n = 4$; Supplementary Fig. 7c, $n = 2$; Supplementary Fig. 7d, $n = 4$; Supplementary Fig. 7e, $n = 2$.

**Reporting summary.** Further information on research design is available in the Nature Research Reporting Summary linked to this article.

## Data availability

The data supporting the findings of this study are available in the article and its supplementary information. Source data are provided with this paper. Original

immunoblot images and plot data generated in this study are provided in the Source Data file. Sequencing datasets have been deposited at the NCBI Sequence Read Archive under the accession code PRJNA750901 and are publicly available. The corresponding processed data are provided in Supplementary Data 1. Human genome 19 (hg19) is publicly available through University of California Santa Cruz [https://genome.ucsc.edu/index.html]. Data are also available from the corresponding authors upon reasonable request. Source data are provided with this paper.

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

## Acknowledgements

The authors are grateful to Mina Pellegrini and Monika Hanf for technical support, Matthias Meyer-Bender for bioinformatics support, and all other members of the Jae lab for helpful discussions. We thank Stefan Krebs, Alexander Graf, and Helmut Blum for next-generation sequencing, Christophe Jung for support with confocal microscopy, and Joshua Kie for support with flow cytometry applications. Portions of this research were conducted in conjunction with the Gene Center genomics core facility (LAFUGA), Flow Cytometry Facility at the Gene Center and Core Facility Bioimaging of the Biomedical Centre of the Ludwig-Maximilians-University Munich. European Research Council Starting Grant 804281 (SOLID) to L.T.J.

## Author contributions

Conceptualization: E.F. and L.T.J.; Investigation: E.F., L.K., and L.T.J.; Writing—Original Draft: L.T.J.; Writing—Reviewing & Editing: E.F., L.K., and L.T.J.; Supervision: E.F. and L.T.J.; Funding Acquisition: L.T.J.

## Funding

## Competing interests

The authors declare no competing interests.
