## [Peer Review File · Nature Communications]

DELE1 tracks perturbed protein import and processing in human mitochondriaREVIEWER COMMENTS

Reviewer #1 (Remarks to the Author):

This interesting manuscript explores the mechanism by which DELE1 relays mitochondrial stress, thus contributing to dissect a still enigmatic process in mammalian cells. Previously, the authors have demonstrated that the short-lived protein DELE1, which normally localizes to the mitochondria, is cleaved into a shorter fragment (S-DELE1) by the stress-activated mitochondrial peptidase OMA1 and released from the mitochondria to the cytosol where it binds and activates HRI, leading to the activation of integrated stress response (ISR). How exactly S-DELE1 leaves the mitochondria was unclear. Moreover, under which conditions this pathway is activated remained to be fully determined.

In this manuscript, the authors dissect how DELE1 is sorted in the mitochondria, the role of OMA1 cleavage for cytosolic release and most importantly discover that DELE1 senses stress caused by defective import pathways or compromised presequence processing. The model that emerges is that DELE1 can sense two types of mitochondrial stresses. One the one side, in presence of defective import non-imported cytosolic DELE1 can trigger the ISR. On the other hand, newly synthesized DELE1 can be cleaved in the intermembrane space on its way to the matrix by the stress-activated protease OMA1, leading to its cytosolic release. A series of elegant experiments shed light on how DELE1 is imported in mitochondria, where the cleavage occurs and the regulation of this cleavage by intrinsic elements in the DELE1 presequence.

The manuscript is convincing, the experiments are solid and the implications of this study shed new light on how mammalian mitochondria communicate with the rest of the cells upon dysfunction. A few questions remain on how exactly DELE1 would sense mitochondrial perturbations during import.

Specific comments:

1) Lines 281-283: The conclusion “These observations suggest that, unlike the Su9 MTS, the endogenous MTS of DELE1 may counteract DELE1 release from depolarized mitochondria..” does not seem justified from the experiments presented in Fig. 3. In fact it is the Su9 MTS that counteract OMA1 cleavage in this context. Moreover, the Su9 MTS is a very strong and effective MTS. It seems that accelerating the rate of import into the matrix renders DELE1 less available for OMA1 cleavage. Is the rate of import the decisive factor here?

2) How much DELE1 is present under steady-state conditions outside the mitochondria? In Figure 2l one can see a reduction of the levels of DELE1 in isotonic samples treated with prot K.

3) The authors show that DELE1 is cleaved during import by OMA1 in the IMM. What remains unclear is how DELE1 is a sensor of mitochondrial import stress. In the case of a block in the import (for example TOM40 depleted cells) non imported cytosolic DELE1 is sufficient to induce CHOP. This is clearly shown and suggests that OMA1 cleavage is in fact dispensable for DELE1 activation of the ISR. How does DELE1 during import work as a sensor? This remains enigmatic. DELE1 is not released unless cleaved by OMA1, raising the question whether DELE1 is really the sensor in this case or rather the signal released from the mitochondria (and OMA1 the sensor).

The interesting experiments shown in Figure 6 do not address if in the case of siRNA for PITRM1 DELE1 release is dependent on OMA1. If yes, as expected, how is the membrane potential of the mitochondria in this case? It would be interesting to at least discuss which other genes encoding mitochondrial proteins showed the same behavior of PITRM1 in the screen.

4) Experiments with Cox5-mCherry or Cox8mCherry are missing a negative control.

5) On page 6 line 243, the authors refer to Fig. 2I. It is not clear why they refer to this experiment here.

6) Table S1 is not indicated in the text. On the journal website there is the wrong upload twice of what should be Table S1.

Reviewer #2 (Remarks to the Author):

This exciting manuscript from the Jae lab investigates the regulation of the mitochondrial stress sensor DELE1 in human cells. In previous publications, DELE1 was uncovered by the Jae and Kampmann labs as an important factor that relays mitochondrial stress signals to the integrated stress response upon cleavage by the mitochondrial protease OMA1 and subsequent release into the cytosol. Here, the authors utilize screens in haploid mammalian cells to identify factors that control DELE1 levels. Their experiments turn up several factors, including genes involved in mitochondrial presequence processing, that play an important role in DELE1 regulation. Based on these findings, Jae and colleagues answer key questions relating to the import and processing of DELE1, showing that it traverses both the OM and IM, and that its cleavage by OMA1 and subsequent release occurs while the protein is in the act of import, allowing the protein to sense alterations in mitochondrial protein processing and import. They also outline an important role for DELE1 in supporting the health of cells that lack key components of mitochondrial protein processing, highlighting a potential important link between this system and neurodegenerative disease.

Overall, this is an exciting and well executed study that provides interesting results that continue the important characterization of the recently identified DELE1. I had reviewed a previous version of this

manuscript for another journal and the authors did a nice job addressing all of the concerns I had raised before, so I do not have many additional items. However, I do think one point of the manuscript that needs clarification is what form of DELE1 (long or short) is activating HRI and CHOP in the context of OM import impairment, i.e. TOM40 depletion. While it is clear from the data that TOM40 depletion activates CHOP in a DELE1-dependent manner, this is still pretty strongly dependent on OMA1 (Figure S5e). Does this suggest that there is OMA1 processing of DELE1 into its short form in TOM40 depleted cells, or is the partial requirement for OMA1 in the context of TOM40 depletion caused by something else? To help clarify this point, it would be great if the authors could include westerns showing the processing of DELE1 in WT and OMA1 deletion cells in the presence and absence of TOM40 depletion. If there is in fact OMA1-dependent processing of DELE1 in TOM40-depleted cells, this could suggest that both the short and full length form contribute to HRI and CHOP activation upon OM import stress. Either way seems okay and interesting, but it is worth knowing the full story under OM import stress.

Reviewer #3 (Remarks to the Author):

In their manuscript „DELE1 is a sensor of perturbed import and processing in human mitochondria”, Fessler and colleagues describe the mechanisms of how DELE1 the shuttling between mitochondria and the cytoplasm is regulated and also utilized to communicate mitochondrial protein import and processing stress to the cell. The authors demonstrate in a set of experiments, that DELE1 employs the mitochondrial presequence import pathway and, in unperturbed conditions, localizes to the matrix. The authors further propose that while going through the presequence pathway itself, DELE1 can sense different types of protein import stress, including import as well as processing defects. Finally, the authors suggest a potential role for DELE1 in human mitochondrial diseases. Overall, the study of Fessler and colleagues adds important knowledge to our current understanding of how mitochondria can communicate stress to the cell.

Generally, the mitochondrial import, processing and localization of the different DELE1 variants L-, M- and S- DELE1 has been well described and substantiated with various different experiments. The same hold true for the importance of the N-terminal part of DELE1 in counteracting its release in steady state conditions. The work might be improved to add information about the mechanism and/or machineries by which DELE1 gets released from mitochondria, as well as how exactly the N-terminal region of the protein favors its retention within the mitochondria remains elusive. The identification of additional factors that bind to DELE1 throughout its processing might help to elucidate how the protein gets retained or released.

Major comments:

1) The band pattern for DELE1 varies between different experiments. Although in their paper describing the identification of DELE1 (<https://doi.org/10.1038/s41586-020-2078-2>) the authors show 3 bands for DELE1 also in mitochondrial isolations, this is not the case in this manuscript (e.g., F1j or F3a). Is this related to exposure times? According to the described mechanism of import, especially L- and M-DELE1 should both appear in the mitochondrial fraction.

2) Along a similar line, according to the deep mutagenesis screen, CLUH should affect whole cell DELE1 level. Therefore, it would be informative to see also whole cell DELE1 level and not only those from isolated mitochondria in F1i. In line with that, it would be good to show DELE1 in F1j, too. As indicated above, it is unclear why in the mitochondrial lysates, there is only one band visible for DELE1.

3) Were the exposers used to detect cleaved and cytoplasmic S-DELE by IF in F4C and E different? From the IF one would still think that following TEV-mediated L-DELE cleavage a considerable amount of the protein gets released from mitochondria. Yet, as also stated by the authors, this is not reflected in the level of CHOP activation. Would it help here to perform a cell fractionation and directly compare cytoplasmic S-DELE level following cleavage with either TEV or CP?

Minor comments:

1) Upon fusion of DELE1 to Su9, the authors observed increased matrix accumulation of M-DELE and decreased cytoplasmic release upon stress induction. Does this affect the down-stream signaling and the activation of HRI?

2) Significances could be added on bar graphs (e.g., F1h and k and other figures).

3) The highest lane in SF1b (DELE1 blot) is lacks quality and should be exchanged.

4) The lane for L- and M-DELE1 in the westernblots shift with respect to the marker annotation throughout the paper (e.g., comparing F3b and g where it seems that L- and M-DELE runs below the 65 kD mark in b and above that mark in g). The authors could check for those small incongruities and double check the marker heights.

REVIEWER COMMENTS

Reviewer #1 (Remarks to the Author):

This interesting manuscript explores the mechanism by which DELE1 relays mitochondrial stress, thus contributing to dissect a still enigmatic process in mammalian cells. Previously, the authors have demonstrated that the short-lived protein DELE1, which normally localizes to the mitochondria, is cleaved into a shorter fragment (S-DELE1) by the stress-activated mitochondrial peptidase OMA1 and released from the mitochondria to the cytosol where it binds and activates HRI, leading to the activation of integrated stress response (ISR). How exactly S-DELE1 leaves the mitochondria was unclear. Moreover, under which conditions this pathway is activated remained to be fully determined.

In this manuscript, the authors dissect how DELE1 is sorted in the mitochondria, the role of OMA1 cleavage for cytosolic release and most importantly discover that DELE1 senses stress caused by defective import pathways or compromised presequence processing. The model that emerges is that DELE1 can sense two types of mitochondrial stresses. One the one side, in presence of defective import non-imported cytosolic DELE1 can trigger the ISR. On the other hand, newly synthesized DELE1 can be cleaved in the intermembrane space on its way to the matrix by the stress-activated protease OMA1, leading to its cytosolic release. A series of elegant experiments shed light on how DELE1 is imported in mitochondria, where the cleavage occurs and the regulation of this cleavage by intrinsic elements in the DELE1 presequence.

The manuscript is convincing, the experiments are solid and the implications of this study shed new light on how mammalian mitochondria communicate with the rest of the cells upon dysfunction. A few questions remain on how exactly DELE1 would sense mitochondrial perturbations during import.

We thank the reviewer for their insightful comments, constructive feedback and for considering our findings to be interesting and important and our experimental strategy to be elegant, convincing and solid.

Specific comments:

1) Lines 281-283: The conclusion “These observations suggest that, unlike the Su9 MTS, the endogenous MTS of DELE1 may counteract DELE1 release from depolarized mitochondria..” does not seem justified from the experiments presented in Fig. 3. In fact it is the Su9 MTS that counteract OMA1 cleavage in this context. Moreover, the Su9 MTS is a very strong and effective MTS. It seems that accelerating the rate of import into the matrix renders DELE1 less available for OMA1 cleavage. Is the rate of import the decisive factor here?

We thank the reviewer for this comment. Firstly, please note that all chimeric constructs shown in the corresponding figure – including Su9-DELE1 – are still cleaved in depolarized (i.e. CCCP-treated) mitochondria (Fig. 3e, g). However, we agree with the reviewer that the Su9-MTS is less permissive for OMA1-mediated cleavage when mitochondria are depolarized. Most importantly, unlike WT-DELE1 and MICU1-DELE1, Su9-DELE1 is no longer cleaved when mitochondria are perturbed with oligomycin (OM). We have investigated these aspects in further detail.

Following the reviewer’s suggestion, we have ‘strengthened’ the identified DELE1 presequence to allow more efficient import by replacing T24 by an arginine residue, based on a similar strategy recently reported for *C. elegans* ATFS-1 (Shpilka *et al.*, Nat Commun., 2021, <https://doi.org/10.1038/s41467-020-20784-y>). While this clearly promoted conversion into M-DELE1, it was still unable to prevent cleavage of DELE1(T24R) in the context of treatment with CCCP or OM (**new Supplementary Fig. 5c**). This indicates, that not just the efficiency of the MTS but more complex mechanisms are at play.

In our revised manuscript, we have uncovered the underlying reason. In the chimeric proteins in Fig.3a the entire ~100aa DELE1 MTS is swapped for that of Su9. This removes parts of the DELE1 MTS which we reveal to be important for cleavability by OMA1 in Fig.5, including a critical region between amino acids 80 and 106, which we identify to strongly promote cleavage by OMA1 in the context of ATP synthase inhibition (please see our response to reviewer #3 below for further details). When we instead replace only the identified *presequence* of DELE1 with that of Su9, we find that the chimera is now readily cleavable after OM treatment (**new Fig. 5e** and **new Supplementary Fig. 5b**). Retaining the presequence-adjacent portion of the DELE1 MTS also seems to benefit cleavage of Su9-DELE1 chimeras in depolarized mitochondria (**new Figure 5e**: compare Su9-DELE1(Δ N36) vs. Su9-DELE1(Δ N106) treated with CCCP). Taken together, our new data demonstrate that, more importantly than the strength of the presequence, the presence of the extended DELE1 MTS facilitates cleavage by OMA1 when matrix import is complicated by ATP synthase malfunction or in the context of a depolarized IMM. As we show in Fig. 5, this is caused by the DELE1 MTS promoting association with stressed mitochondria, allowing OMA1 to cleave. As this mechanism is only revealed in Fig.5, we agree with the reviewer that the conclusion would be premature at the end of Fig. 3. We have rephrased the text at the respective section accordingly.

2) How much DELE1 is present under steady-state conditions outside the mitochondria? In Figure 2l one can see a reduction of the levels of DELE1 in isotonic samples treated with prot K.

In our system, working with endogenously tagged DELE1 or exogenously expressed DELE1 at very modest levels, we do not detect DELE1 in the cytosolic fraction in unstressed cells. However, we indeed expect an amount of DELE1 in the process of mitochondrial import to still have the C-terminus exposed to the cytosol. As requested, we have quantified the decrease in DELE1 signal under the different buffer conditions (isotonic, hypotonic, TX-100 lysed) and include these data in **new Supplementary Fig. 2d**. This shows that, in line with our other observations, by far the most profound decline is observed when the matrix is rendered accessible with TX-100, but some reduction in signal occurs also in isotonic and hypotonic buffer upon addition of proteinase K. Strikingly, as mentioned above, in our revised manuscript, we show that inhibition of the ATP synthase with OM dramatically increases sensitivity of DELE1 – including M-DELE1 – to proteinase K in intact mitochondria, which leads us to identify a critical region in the extended DELE1 MTS that enables cleavage by OMA1 in this setting of mitochondrial perturbation (please see below our comments to reviewer #3 for further details).

3) The authors show that DELE1 is cleaved during import by OMA1 in the IMM. What remains unclear is how DELE1 is a sensor of mitochondrial import stress. In the case of a block in the import (for example TOM40 depleted cells) non imported cytosolic DELE1 is sufficient to induce CHOP. This is clearly shown and suggests that OMA1 cleavage is in fact dispensable for DELE1 activation of the ISR. How does DELE1 during import work as a sensor? This remains enigmatic. DELE1 is not released unless cleaved by OMA1, raising the question whether DELE1 is really the sensor in this case or rather the signal released from the mitochondria (and OMA1 the sensor).

Based on our previous data, one possibility could have been that *only* S-DELE1 is able to bind to and activate HRI, for example, because the presence of the MTS in L-DELE1 could have prevented the interaction with the kinase. Our new results clearly demonstrate that this is not the case and that, as the reviewer points out, scenarios can be found in which mitochondrial import stress can be sensed and signaled by DELE1 independently of OMA1, in a manner that is reminiscent of ATFS-1 in *C. elegans*. Besides the mentioned perturbation of TOM40, our new data generated in response to the reviewer's suggestions show that this can, for instance, also be elicited by PITRM1 loss (**new Supplementary Fig. 7b**), and we anticipate that additional settings will be identified in the future.

At the same time, we acknowledge the important role of OMA1 in many scenarios. Notably, in cases where OMA1 is required, it needs to collaborate with DELE1 in order for the stress sensing

to be productive. In our view, this positions DELE1 as a critical and active puzzle piece in mitochondrial stress sensing processes. Following the reviewer's comment, we have changed the title and reworded the text throughout and no longer refer to DELE1 as a 'sensor'.

The interesting experiments shown in Figure 6 do not address if in the case of siRNA for PITRM1 DELE1 release is dependent on OMA1. If yes, as expected, how is the membrane potential of the mitochondria in this case?

We thank the reviewer for bringing this question to our attention. We have performed the requested experiments and present the results in **new Supplementary Figs. 7a, b**. Interestingly, as mentioned above, this also reveals that in the absence of OMA1 activity, PITRM1 depletion leads to a cytosolic association of L-DELE1 with HRI, again highlighting the multi-faceted nature of DELE1's response to mitochondrial import stress. Regarding the membrane potential, we observe a decrease in TMRM signal in cells depleted of PITRM1 (**additional Fig. 1**). At the same time, we envision the consequences of PITRM1 deficiency to be complex and would like to underline that OMA1 can also be triggered in settings where the mitochondrial membrane potential is not lost (such as ATP synthase inhibition by oligomycin).

Additional Figure. 1.

PITRM1 was silenced in HeLa cells by two different shRNAs for 72h and TMRM signal was measured by flow cytometry. Graph depicts mean \pm s.d. of n=3 independent experiments (ordinary one-way ANOVA with Dunnett's multiple comparisons correction).

It would be interesting to at least discuss which other genes encoding mitochondrial proteins showed the same behavior of PITRM1 in the screen.

We now highlight all other significant mitochondrial proteases in Fig.1f and mention them in the text. Furthermore, we validate an additional hit among those, CLPP, which scores like PITRM1 in the screen (**new Supplementary Fig. 7e**).

4) Experiments with Cox5-mCherry or Cox8mCherry are missing a negative control.

We apologize for this omission and have included the control (**new Fig. 6b**).

5) On page 6 line 243, the authors refer to Fig. 2l. It is not clear why they refer to this experiment here.

The confusing referral has been removed.

6) Table S1 is not indicated in the text. On the journal website there is the wrong upload twice of what should be Table S1.

In the revised manuscript Table S1 has been added to the text alongside the first mention of the corresponding Fig. 1f. We have also double-checked the submitted Tables S1 and S2 in the electronic submission system and could not find any issue with them on our end. We will be sure to check this again prior to publication.

Reviewer #2 (Remarks to the Author):

This exciting manuscript from the Jae lab investigates the regulation of the mitochondrial stress sensor DELE1 in human cells. In previous publications, DELE1 was uncovered by the Jae and Kampmann labs as an important factor that relays mitochondrial stress signals to the integrated stress response upon cleavage by the mitochondrial protease OMA1 and subsequent release into the cytosol. Here, the authors utilize screens in haploid mammalian cells to identify factors that control DELE1 levels. Their experiments turn up several factors, including genes involved in mitochondrial presequence processing, that play an important role in DELE1 regulation. Based on these findings, Jae and colleagues answer key questions relating to the import and processing of DELE1, showing that it traverses both the OM and IM, and that its cleavage by OMA1 and subsequent release occurs while the protein is in the act of import, allowing the protein to sense alterations in mitochondrial protein processing and import. They also outline an important role for DELE1 in supporting the health of cells that lack key components of mitochondrial protein processing, highlighting a potential important link between this system and neurodegenerative disease.

Overall, this is an exciting and well executed study that provides interesting results that continue the important characterization of the recently identified DELE1. I had reviewed a previous version of this manuscript for another journal and the authors did a nice job addressing all of the concerns I had raised before, so I do not have many additional items. However, I do think one point of the manuscript that needs clarification is what form of DELE1 (long or short) is activating HRI and CHOP in the context of OM import impairment, i.e. TOM40 depletion. While it is clear from the data that TOM40 depletion activates CHOP in a DELE1-dependent manner, this is still pretty strongly dependent on OMA1 (Figure S5e). Does this suggest that there is OMA1 processing of DELE1 into its short form in TOM40 depleted cells, or is the partial requirement for OMA1 in the context of TOM40 depletion caused by something else? To help clarify this point, it would be great if the authors could include westerns showing the processing of DELE1 in WT and OMA1 deletion cells in the presence and absence of TOM40 depletion. If there is in fact OMA1-dependent processing of DELE1 in TOM40-depleted cells, this could suggest that both the short and full length form contribute to HRI and CHOP activation upon OM import stress. Either way seems okay and interesting, but it is worth knowing the full story under OM import stress.

We thank the reviewer for their constructive feedback and helpful suggestions. We appreciate that the reviewer finds our work exciting, well-executed and important and acknowledges the potential important link to human neurodegenerative disease.

Whereas different steps of matrix import at the level of the TIM complex can be readily inhibited with chemicals like CCCP and OM, manipulation of the TOM complex is less explored. We would expect that a complete block of TOM40-mediated import is unlikely and some background activity remains in place. Following the reviewer's suggestion, we have investigated DELE1 cleavage by OMA1 in cells subjected to TOM40 silencing and indeed detect that some amount of S-DELE1 can be generated in this setting (**new Supplementary Fig. 6e**). Moreover, we have performed the cytosolic interaction assay by digitonin permeabilization. In line with some amount of S-DELE1 being produced, and as predicted by the reviewer, this indeed shows that HRI can co-precipitate both L- as well as S-DELE1 in the context of TOM40 silencing (**new Supplementary Fig. 6j**), whereas only L-DELE1 is co-precipitated in the absence of OMA1 (Supplementary Fig. 6k). Of note, we find the cell's ability to induce CHOP upon TOM40 inhibition to require DELE1 (Supplementary Fig. 6d) but not OMA1 (**new Supplementary Fig. 6e**).

Reviewer #3 (Remarks to the Author):

In their manuscript „DELE1 is a sensor of perturbed import and processing in human mitochondria”, Fessler and colleagues describe the mechanisms of how DELE1 the shuttling between mitochondria and the cytoplasm is regulated and also utilized to communicate mitochondrial protein import and processing stress to the cell. The authors demonstrate in a set of experiments, that DELE1 employs the mitochondrial presequence import pathway and, in unperturbed conditions, localizes to the matrix. The authors further propose that while going through the presequence pathway itself, DELE1 can sense different types of protein import stress, including import as well as processing defects. Finally, the authors suggest a potential role for DELE1 in human mitochondrial diseases. Overall, the study of Fessler and colleagues adds important knowledge to our current understanding of how mitochondria can communicate stress to the cell.

Generally, the mitochondrial import, processing and localization of the different DELE1 variants L-, M- and S- DELE1 has been well described and substantiated with various different experiments. The same hold true for the importance of the N-terminal part of DELE1 in counteracting its release in steady state conditions. The work might be improved to add information about the mechanism and/or machineries by which DELE1 gets released from mitochondria, as well as how exactly the N-terminal region of the protein favors its retention within the mitochondria remains elusive. The identification of additional factors that bind to DELE1 throughout its processing might help to elucidate how the protein gets retained or released.

We thank the reviewer for their helpful comments and for recognizing the important knowledge on mitochondrial stress signaling added by our study. Following the reviewer’s comments and related to a question raised by reviewer #1, we have investigated the biology of the DELE1 N-terminal region in further detail. Firstly, as detailed in our response to reviewer #1 (please see above), we find that the strength of the presequence is overshadowed by a more complicated mechanism involving the extended DELE1 MTS (**new Fig. 5e** and **new Supplementary Figs. 5b, c**). Next, when the IMM is depolarized (and DELE1 translocation across the IMM is thus blocked), the DELE1 presequence can expectably not be severed by MPP, which leads to the observed accumulation of L-DELE1 in the absence of OMA1 (Fig. 2b; Supplementary Fig. 3a) outside of the mitochondrial matrix, where it is sensitive to proteinase K (Fig. 3a). There, extended parts of DELE1 upstream of the OMA1 cleavage site hinder its release from mitochondria (Fig. 5a-c). By contrast, when the ATP synthase is perturbed and the initial steps of matrix import (i.e. presequence translocation and cleavage) are intact, we find that M-DELE1 is still generated (Fig. 2a, b), as would be expected. Strikingly, however, in this setting, the C-terminus of many M-DELE1 molecules is readily accessible to proteinase K from the cytosol, indicating an import arrest in a $N_{in}-C_{out}$ configuration (**new Figure 5f**). This type of arrest can be observed for multiple mitochondrial proteins in the context of perturbed matrix import and often involves hydrophobic regions acting as ‘stop-transfer’ signals. Hydrophobicity analysis reveals a short stretch of amino acids at the end of the extended DELE1 MTS, which is computationally predicted to adopt an alpha-helical configuration. Mutant analysis demonstrates that this motif is the critical region enabling DELE1 cleavage in mitochondria with perturbed ATP synthase activity (**new Fig. 5d, g** and **new Supplementary Fig. 5d**). Remarkably, over the past years, a similar behavior has also been pieced together for the sorting of another key mitochondrial stress sensor, PINK1 (Sekine & Youle, BMC Biol., 2018, <https://doi.org/10.1186/s12915-017-0470-7>).

Major comments:

1) The band pattern for DELE1 varies between different experiments. Although in their paper describing the identification of DELE1 (<https://doi.org/10.1038/s41586-020-2078-2>) the authors show 3 bands for DELE1 also in mitochondrial isolations, this is not the case in this manuscript (e.g., F1j or F3a). Is this related to exposure times? According to the described mechanism of import, especially L- and M-DELE1 should both appear in the mitochondrial fraction.

We understand the confusion, but this can readily be resolved. The relevant panel in Fig.1 depicts endogenous DELE1^{Alfa} signal in the mitochondrial fraction of HAP1 cells. Besides the fact that endogenous DELE1 expression is very low in HAP1 cells, in our hands, the reagents for the detection of Alfa-tagged proteins perform excellently for native conditions (like FACS), but are poor for detection in immunoblotting. Despite these challenges, we think that L- and M-DELE1^{Alfa} are still visible in wild-type cells (first lane) in Fig.1i, of which we provide an enlarged version below (**additional Fig. 2**). In CLUH-KO cells (last lane), DELE1 signal is expectedly much reduced. In PITRM1-KO cells (middle lanes), M-DELE1 disappears, in line with our other data presented in the manuscript, and rationalized by the notion that PITRM1 inhibition causes a backlog that impairs precursor processing by MPP (Kücükköse *et al.*, FEBS J., 2021, <https://doi.org/10.1111/febs.15358>). To aid visibility of M-DELE1 despite the technical challenges in this particular setting, we have scaled up the mitochondrial isolation from HAP1 cells and provide a new blot for Fig.1i, in which the two bands (L- and M-DELE1) should be more readily visible in wild-type cells (**new Fig. 1i**).

Our main focus in Fig. 3a was the differential sensitivity of DELE1 to proteinase K upon CCCP treatment. In this case, M-DELE1 is not present anymore, as matrix import is inhibited. As the clarity of the DELE1 bands in the DMSO-treated samples was suboptimal, we have improved our experimental set-up and provide a new figure in which L- and M-DELE1 should be more readily visible and have been labelled with the respective colored dots (**new Fig. 3a**).

Additional Figure. 2.

Endogenous DELE1^{Alfa} in mitochondria isolated from HAP1 DELE1^{Alfa} cells detected by immunoblotting using FluoTag®-X2 anti-ALFA for Western Blotting.

2) Along a similar line, according to the deep mutagenesis screen, CLUH should affect whole cell DELE1 level. Therefore, it would be informative to see also whole cell DELE1 level and not only those from isolated mitochondria in F1i. In line with that, it would be good to show DELE1 in F1j, too. As indicated above, it is unclear why in the mitochondrial lysates, there is only one band visible for DELE1.

For the reasons explained above, in our hands, endogenous DELE1^{Alfa} is too diluted in HAP1 whole cell lysates for detection by immunoblotting (Fig. 1j). However, the effect of CLUH on whole cell DELE1 levels can be seen in Fig. 1h (showing DELE1 signal in flow cytometry of whole cells). Additionally, following the reviewer's suggestion, we now also provide new immunoblot data showing endogenous DELE1^{HA} in whole cell lysates of HeLa cells exposed to sgRNAs directed against CLUH (**new Supplementary Fig. 1b**).

3) Were the exposers used to detect cleaved and cytoplasmic S-DELE by IF in F4C and E different? From the IF one would still think that following TEV-mediated L-DELE cleavage a considerable amount of the protein gets released from mitochondria. Yet, as also stated by the

authors, this is not reflected in the level of CHOP activation. Would it help here to perform a cell fractionation and directly compare cytoplasmic S-DELE level following cleavage with either TEV or CP?

We thank the reviewer for this observation. Although TOM20 only represents a non-essential control in this experiment (the relevant comparison being between TIM50 and Su9), indeed, small amounts of cytosolic DELE1(TCS) can be observed in some cells treated with TOM20-TEV. This is also visible in the fractionation, which we have performed following the reviewer's suggestion (**additional Fig. 3**). However, the amounts produced pale in comparison to what we observe with 3C protease and, most importantly, the occurrence of S-DELE1 generated by TOM20-TEV does not correspond to the observed reduction in the L- and M-DELE1, which can also be seen in former Fig. 4a. This points to an additional technical limitation of TEV protease in this setting, possibly caused by TEV cleavage generating a destabilizing motif rendering the cleavage product unstable in the cytosol. Given that this is a technical limitation of TEV protease unrelated to the studied biology, and that we have long found a robust solution in 3C protease, we decided to remove the TEV data and instead elaborate on the 3C protease data by showing that no induction of CHOP is triggered when DELE1 is lacking the 3C site or when catalytic-dead versions of 3C protease are used (**new Supplementary Fig. 4b**).

Additional Figure 3.

Fractionation in OMA1-deficient HeLa cells upon transfection of DELE1(TCS) and the indicated TEV constructs.

Minor comments:

1) Upon fusion of DELE1 to Su9, the authors observed increased matrix accumulation of M-DELE and decreased cytoplasmic release upon stress induction. Does this affect the downstream signaling and the activation of HRI?

We have tested the downstream signaling of these synthetic constructs and provide the results in **new Supplementary Fig. 3c**, showing that the ability of Su9-DELE1 to activate CHOP upon OM treatment is blunted.

2) Significances could be added on bar graphs (e.g., F1h and k and other figures).

We have added significances throughout where sensible.

3) The highest lane in SF1b (DELE1 blot) is lacks quality and should be exchanged.

As explained above, the lower quality is caused by the technical challenges of detecting endogenous DELE1^{Alfa} in HAP1 cells by immunoblotting. We have replaced the picture for a better exposure as requested. As mentioned above, a superior immunoblot in clonal knockout cells is now shown in **new Fig. 1i**.

4) The lane for L- and M-DELE1 in the westernblots shift with respect to the marker annotation throughout the paper (e.g., comparing F3b and g where it seems that L- and M-DELE runs below the 65 kD mark in b and above that mark in g). The authors could check for those small incongruities and double check the marker heights.

We are thankful for the reviewer's keen eye. However, this does not reflect incongruities but stems from the fact that the cDNA expression vector for DELE1 contains a longer linker sequence upstream of the HA-tag compared to the construct that we used to target the endogenous DELE1 locus with a C-terminal HA-tag. For clarity, we have taken care to always explicitly identify endogenously edited cells by italicizing the edited locus (e.g. HeLa *DELE1^{HA}*) and differentiate it from cDNAs (e.g. DELE1-HA) throughout the figures and legends.

REVIEWERS' COMMENTS

Reviewer #1 (Remarks to the Author):

In this revised version, the authors have performed additional experiments that answer and clarify the raised points. The manuscript in the present form presents a step forward to understand how DELE1 integrates mitochondrial stress responses upon different perturbations.

Reviewer #2 (Remarks to the Author):

The authors convincingly addressed the issues I raised in my previous review, and I congratulate them on an exciting study.

Reviewer #3 (Remarks to the Author):

Fessler and colleagues have thoroughly and in a very detailed manner addressed all comments and concerns raised during the review process and added new, conclusive experiments to the manuscript where needed.

REVIEWERS' COMMENTS

Reviewer #1 (Remarks to the Author):

In this revised version, the authors have performed additional experiments that answer and clarify the raised points. The manuscript in the present form presents a step forward to understand how DELE1 integrates mitochondrial stress responses upon different perturbations.

Reviewer #2 (Remarks to the Author):

The authors convincingly addressed the issues I raised in my previous review, and I congratulate them on an exciting study.

Reviewer #3 (Remarks to the Author):

Fessler and colleagues have thoroughly and in a very detailed manner addressed all comments and concerns raised during the review process and added new, conclusive experiments to the manuscript where needed.

We once again thank all reviewers for evaluating our manuscript and their constructive feedback.